# Diamond growth from organic compounds in hydrous fluids deep within the Earth

Maria Luce Frezzotti [1]*

At subduction zones, most diamonds form by carbon saturation in hydrous fluids released from lithospheric plates on equilibration with mantle rocks. Although organic molecules are predicted among dissolved species which are the source for carbon in diamonds, their occurrence is not demonstrated in nature, and the physical model for crustal diamond formation is debated. Here, using Raman microspectroscopy, I determine the structure of carbon-based phases inside fluid inclusions in diamond-bearing rocks from the Alps. The results provide direct evidence that diamond surfaces are coated by $sp^2$-, and $sp^3$-bonded amorphous carbon and functional groups of carboxylic acids (e.g., carboxyl, carboxylate, methyl, and methylene), indicating the geosynthesis of organic compounds in deep hydrous fluids. Moreover, this study suggests diamond nucleation via metastable molecular precursors. As a possible scenario, with carbon saturation by reduction of carboxylate groups, I consider tetrahedral H-terminated C groups as templates for the growth of $sp^3$-structured carbon.

[1] Dipartimento di Scienze dell'Ambiente e della Terra, Università Milano Bicocca, Piazza della Scienza 1-U4, 20126 Milano, Italy. *email: maria.frezzotti@unimib.it

Diamonds account for tiny amounts of carbon from the Earth's interior[1,2], yet revealing their genesis is necessary for understanding large-scale processes in the evolution of our planet, such as geodynamics, volatile cycling, and redox state[3–6]. At subduction zones, where dense lithospheric plates sink to asthenospheric depths, diamond formation and carbon cycling are intrinsically related, since carbon fluxes inside the Earth are regulated by oxidized—dominantly hydrous—fluids (including supercritical aqueous-silicate intermediates) delivered by metamorphic decarbonation and dissolution of crustal carbonates[7–10]. In deep C-O-H fluids, the transition from carbon transport to carbon saturation—as diamond—requires equilibrium with slab or mantle rocks, controlled by those parameters that influence the solubility of carbon at high pressure ($P$) and temperature ($T$) (e.g., $P = 3–5$ GPa; $T = 600–900\,°C$), including pH, redox conditions, structure of solutes in fluids, and buffering mineral assemblages in rocks[5,11–15]. The formation of crustal metamorphic microdiamonds, therefore, is primarily governed by the nature of dissolved carbon-bearing ionic and molecular species.

In recent years, seminal work has revealed that $CO_2$ does not regulate total carbon in hydrous fluids[7,9–12] as hitherto generally accepted. High-pressure conditions can change carbon speciation substantially: aqueous carbonate ions can also be present, which would result, substantially, in a higher carbon solubility[9,10]. Further, modeling by thermodynamic calculations (Deep Earth Water model[11]) has recently predicted the rise of stability of organic acids[11,12]; at pressures above 5 GPa, formic acid can dissociate into molecular water and diamond. The question consequently arises if organic structures can be stable in hydrous fluids deep in the Earth and if these can trigger diamond formation. The existence of organic molecules in subduction fluids would bear far-reaching implications for the genesis of diamonds, for carbon speciation in deep fluids, and for the potential for prebiotic compounds in the Earth.

By using Raman microspectroscopy, I studied the structure and composition of carbon-bearing phases in fluid inclusions present in metamorphic rocks from Lago di Cignana[16] (western Alps), where microdiamonds were previously discovered[9]. Compared to experimental or theoretical studies, the analysis of fluid-containing inclusions has the advantage of examining a direct sample of diamond-forming media encapsulated within minerals[17]. Fluid inclusions can retain the natural products, frozen in, on carbon saturation. The results show that significant amounts of carboxylic acids are dissolved in aqueous fluids released from deeply subducting slabs, and how they contribute to the nucleation and growth of diamonds.

## Results

**Diamond-bearing fluid inclusions in metasediments.** The diamond-bearing ultrahigh-pressure (UHP) ophiolitic slice of Lago di Cignana (western Alps) has several characteristics, which are very appropriate to study diamond formation in subduction zones. In brief, the UHP metamorphic complex represents a fragment of an exhumed metamorphic complex of crustal origin subducted to a depth of 100 km, or more, during Alpine oceanic subduction[16] about 35 Ma ago. The host metamorphic rocks are of oceanic affinity, contrarily to the commonly reported continental origin of diamond-bearing UHP metamorphic rocks[18,19]. Diamond crystallization occurred by precipitation from an oxidized hydrous fluid buffered by redox equilibrium with surrounding slab rocks[9,20], which is preserved in fluid inclusions.

The UHP metamorphic complex formed at $P \geq 3.2$ GPa and relatively low $T$ (i.e., 600 °C)[16] and consists of glaucophane eclogites from original oceanic-crust basalts, and a metasedimentary cover series (e.g., quartz-rich micaschists and quartzites)[16,18]. The microdiamonds occur in nodules and boudinaged layers of Mn-rich garnetites within micaschists and quartzites, which likely reflect Mn nodules and metasediments[16,21]. Diamonds are also observed within primary fluid inclusions distributed in garnet cores (Fig. 1a). These inclusions are the focus of the present study. They are tiny (<1–25 μm), dominantly aqueous, two or three-phase inclusions ($L + S \pm V$) (Fig. 1b). The vapor bubble, when present, resulted from a volume increase, due to inclusion stretching on host-rock exhumation (refer to Supplementary Note 1 and Supplementary Fig. 1). In inclusions, fluid-precipitated daughter mineral phases include—along with rare diamond—carbonates, rutile, salts, and quartz (Fig. 1b; Supplementary Table 1). Dissolved carbonate and bicarbonate ions, silica monomers, and chlorides are detected in liquid water[9,20] (see Supplementary Note 1). The relative abundances of dissolved and precipitated carbon species reveal high concentrations of carbon and an oxidized nature of diamond-forming fluids equilibrated with oceanic-crust rocks, which is in agreement with the experimental data on carbon solubility at the considered $P–T$ conditions (dissolved C = 0.1–0.3 wt%)[7].

**Raman study of diamonds.** Under the Raman analysis, most diamonds (≤1–2 μm in size) contained inside inclusions show crystalline structure revealed by the $sp^3$-hybridized mode at 1332 cm$^{-1}$ (ref. [22]), identical to previously studied microdiamonds from this locality[9] (Table 1). In several fluid inclusions, however, diamonds spectra show several additional bands arising from other non-diamond forms of carbon, whose attribution requires deconvolution and band fitting (for a detailed account of the Raman study, refer to Methods and Supplementary Note 2). The Raman spectra of these diamonds are shown in Figs. 2 and 3. Two prominent bands, D and G, located around 1360 and 1580 cm$^{-1}$, are from poorly ordered $sp^2$-bonded carbon[23] (Fig. 2a–f; Table 2). The G band corresponds to the vibration of carbon atoms in $sp^2$ sites, completed by the intense and broad D band originating from distinct modes of disorder in aromatic ring structures[23,24]. Besides, a band around 1520–1560 cm$^{-1}$ (Fig. 2a–d) argues for the presence of an amorphous, probably hydrogenated, carbon fraction in the $sp^2$ and $sp^3$ configurations[24], whereas two bands at around 1160 and 1440 cm$^{-1}$ (Fig. 2c–e) indicate terminations with $trans$-polyacetylene [$trans$-$(CH)_n$] segments[23,25]. A few spectra show the presence of metastable diamond polytypes (sharp peaks in the 1280–1350 cm$^{-1}$ region in Fig. 2e). As shown in Fig. 2, intensity and number of Raman bands reveal a variable $sp^2/sp^3$ carbon ratio. With increasing amorphous and disordered $sp^2$-bonded carbon content (from 0.1 to 8% $sp^2$-bonded C fraction; see Methods and Table 3), the diamond peak is shifted (1334 and 1328 cm$^{-1}$), suppressed, and broadened (full-width at half-peak maximum (FWHM) from 6 to 10.6 cm$^{-1}$) (spectra a to d in Fig. 2; Table 1). These features suggest a diamond size in the order of hundreds of nanometers[26]. In nano-sized (<1 μm) diamond spectra, a broad band centered around 1220–1226 cm$^{-1}$ (spectra a and b in Fig. 2) suggests even smaller diamonds or abundant planar defects[27,28].

None of the analyzed Raman bands of $sp^2$-bonded carbon correspond to metamorphic graphite[23,29,30] (see Supplementary Note 2). The structural analysis of poorly ordered and amorphous carbon coating nano-sized diamonds can be interpreted from fitted bands (Table 2) following the three-stage amorphization-trajectory model of Ferrari and Robertson[23], where the increasing disorder is revealed by the variation of the position of the G band, and by the relative intensity ratio of the D and G bands. The relative integrated intensity ratio $I_D/I_G$ for the D and G band pair

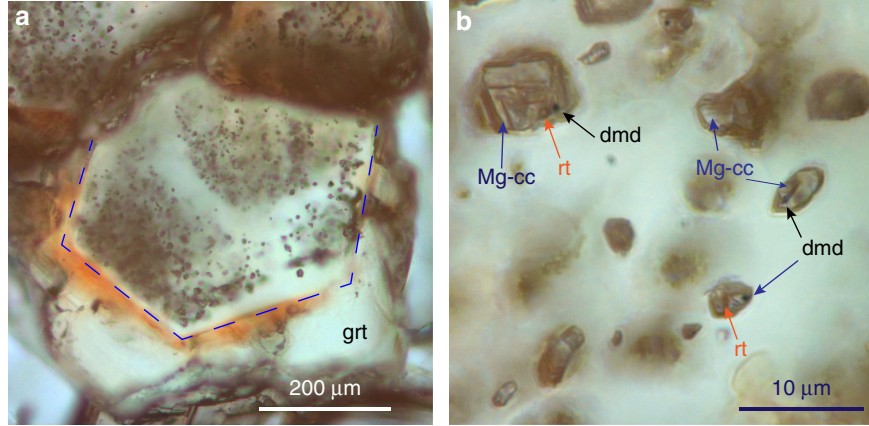

**Fig. 1** Photomicrographs of diamond-bearing fluid inclusions in garnetites from Lago di Cignana (W Alps). **a** Distribution of fluid inclusions in garnet cores (grt; blue dotted line). **b** Aqueous fluid inclusions containing diamond (dmd), Mg-calcite (Mg-cc), and rutile (rt); stacked microphotograph

**Table 1 Raman features of diamonds in fluid inclusions: Raman data of diamonds in garnet for comparison[a]**

| Diamonds in fluid inclusions | | | | | Diamonds in garnet | | | | |
|---|---|---|---|---|---|---|---|---|---|
| Sample no. | FI no. | Diamond ($cm^{-1}$) | FWHM ($cm^{-1}$) | FI size (μm) | Sample no. | FI no. | Diamond ($cm^{-1}$) | FWHM ($cm^{-1}$) | dmd size (μm) |
| D2AG | B1e | 1331 | 9 | 8 | 3C2GC | 1D | 1330.8 | 9.1 | 3 |
| M2G | M5 | 1331 | 9 | 8 | 3C2GC | 3D | 1332 | 6.1 | 2 |
| E2CG | M6 | 1332 | 4 | 10 | E2CG | M3 | 1333 | 7.5 | 2 |
| M2G | A8 | 1333 | 6.1 | 16 | 3C2G | H2 | 1332.5 | 6 | 3 |
| M2G | B2 | 1332 | 4.5 | 5 | 3C2G | H4 | 1331 | 7 | 3 |
| M2G | B6 | 1332 | 4.8 | 7 | 3C2G | H5 | 1331 | 4.5 | 3 |
| M2G | C1 | 1330 | 7.6 | 20 | E2CG | M1 | 1332.5 | 4.5 | 3 |
| M2G | C6 | 1333.7 | 6.1 | 4 | 3C2G | M2 | 1331 | 4 | 4 |
| M2G | C7 | 1332 | 8.4 | 10 | 3C2G | E6 | 1331 | 2.5 | 3 |
| D2AG | A1 | 1328 | 10.6 | 6 | 3C2BG | A1 | 1333 | 7.6 | 4 |
| D2AG | A2 | 1331 | 4 | 6 | 3C2BG | A5 | 1332 | 5 | 5 |
| D2AG | B1 | 1331.5 | 10.6 | 6 | 3C2BG | A6a | 1333 | 4 | 4 |
| G2C | D2 | 1330.8 | 6.1 | 10 | 3C2BGa | 6b | 1333 | 3.5 | 4 |
| G2C | D5 | 1334 | 6.1 | 18 | 3C2G | M2 | 1331 | 6 | 2 |
| G2C | D6 | 1332.4 | 9.1 | 15 | | | | | |
| G2C | E4 | 1331 | 4.6 | 6 | | | | | |
| G2C | E5 | 1328 | 7.6 | 6 | | | | | |
| G2C | E7 | 1330 | 6.1 | 4 | | | | | |
| G2C | E8 | 1331 | 6 | 4 | | | | | |
| G2C | E9 | 1328 | 7.6 | 5 | | | | | |
| ALC2 | B2 | 1328 | 8 | 6 | | | | | |
| ALC2 | B5 | 1329.5 | 8 | 7 | | | | | |
| ALC2 | B6 | 1329 | 8 | 5 | | | | | |
| 3C2G | D5 | 1334 | 6 | 12 | | | | | |
| M2G | C3 | 1333 | 14 | 12 | | | | | |

*FI* fluid inclusion, *FWHM* full-width at half-maximum, *dmd* diamond
[a]Data are from refs. [9,20]

goes from dominantly nano-crystalline disordered graphitic clusters in microdiamonds (Fig. 2f) to nano-crystalline disordered graphitic clusters and amorphous $sp^2$ and $sp^3$ carbon in nano-sized diamonds (Fig. 2b, c).

A few nano-sized diamonds coated by conspicuous disordered and amorphous carbon in the $sp^2$ and $sp^3$ configurations (Fig. 2a–d) show Raman modes of attached functional groups, whose width and featureless nature suggest organic compounds (refer to Supplementary Note 2; Fig. 3; Table 2). Organic functional groups are unevenly recorded in spectra as they do not have fixed sites on the diamond surface. Two bands at around 1724 and 1860 $cm^{-1}$ (Fig. 3a; Table 2) are the signature of the fundamental stretching of the carbonyl group (C=O) in organic acids and acid anhydrides[31]. The presence of methyl $(CH_3)_n$ and

methylene $(CH_2)_n$ radicals is revealed by asymmetric and symmetric C-H stretching bands in the spectral region between 2840 and 2930 $cm^{-1}$ (Fig. 3b; Table 2; refs. [32,33]). Hydration and hydroxylation on the surface have not been observed, since O-H stretching (3200–3800 $cm^{-1}$) and bending (1650 $cm^{-1}$) water bands[34,35] are absent in the diamond spectra (Fig. 3).

**Surface reactivity of diamonds reveals organic species.** Raman analyses provide direct evidence that nano-sized diamonds show hybrid structures, constituted by a diamond core with thin outer domains consisting of non-diamond forms of carbon and organic functional groups (Fig. 3c). To further analyze the results obtained, it is necessary to consider the specific features of the

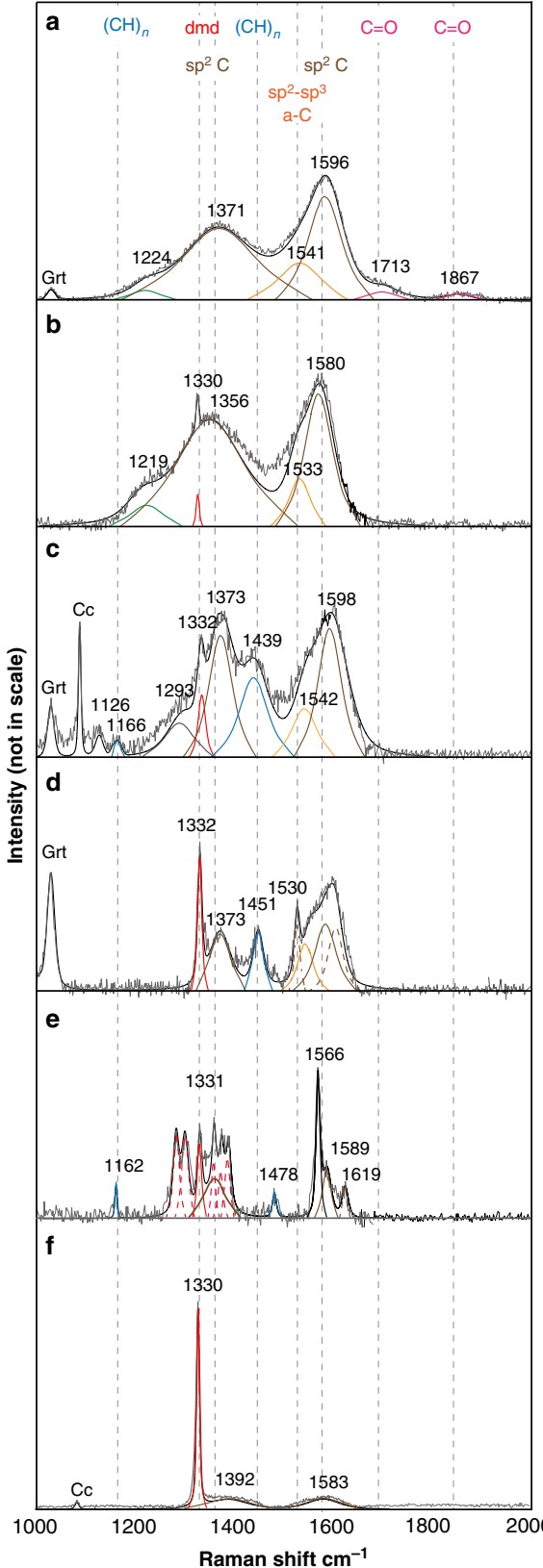

**Fig. 2** Raman spectra showing the structure of carbon-based species in diamonds. **a** A nano-sized diamond with a shell of sp[2] and sp[3] amorphous carbon, as confirmed by the presence of a broad band at 1224 cm[−1] (green band profile). A contribution from surface C=O groups (pink band profiles) is observed. **b** A crystalline nano-sized diamond (peak at 1330 cm[−1]) with amorphous carbon at the surface. **c**, **d** Microdiamonds, with variable through minor contribution from amorphous carbon, show termination with (CH)n segments (blue band profiles). **e** A crystalline microdiamond coexists with other diamond polytypes (red dotted band profiles in the 1300–1350 cm[−1] region). **f** The microdiamond (1 μm) shows almost no contribution from surface sp[2]-bonded carbon. Grt = host garnet; Cc = calcite

carboxyl, anhydride, lactone, methyl, methylene) from the crystallization medium, and spontaneously reconstruct to form sp[2]-bonded amorphous or graphitic carbon shells or clusters, in order to attain equilibrium[36–41]. On the other hand, nanodiamond surface modifications could also result by exposure to a solvent at relatively lower temperature[27,42]. For example, thermal annealing in vacuum, $H_2$, He, or $N_2$, primarily produces sp[2] amorphous (or graphitic) carbon and H-bonded carbon on surfaces, while O-bonded carbon species disappear (e.g., $T > 200$ °C; refs. [43–45]).

Alternatively, oxidation in air, $CO_2$, or aqueous phase increases the content of oxygen-containing functional groups (e.g., O-H, C=O, COOH; $T < 400$ °C), and induces dehydrogenation and removal of sp[2] carbon from the surface[27,41,42,46]. To be consistent with surface oxidation in water at low temperature, the Raman bands of H-bonded carbon groups and nano-crystalline and amorphous sp[2]/sp[3] carbon must show an explicit reduction. However, carboxyl and carboxylate group bands ($\cong$1700–1900 cm[−1]) increase with increasing methyl and methylene radicals ($\cong$2800–3000 cm[1]) and sp[2] (or sp[2]/sp[3])-bonded carbon ($\cong$1300–1600 cm[−1]) (Figs. 2a, 3), thus suggesting that (C=O)-containing surface species cannot result purely from transformation by hydroxyl radicals in aqueous fluids. Moreover, bands arising from O-H groups (e.g., the O-H bend at around 1650 cm[−1]), which form by reaction with or on adsorption from water, are not observed (Figs. 2 and 3a; refs. [34,35]). Thus, diamonds appear to have been stable against $H_2O$ in the fluid inclusions during the retrograde cooling path, and at ambient conditions.

Consequently, based on the present analyses, I conclude that the organic functional groups on the surfaces of Lago di Cignana diamonds were generated by adsorption or reaction with dissolved chemical species in the crystallization fluids at conditions relevant to deep subduction zone environment. The specificity of coexisting O- and H-bonded organic functional groups suggests that they should be part of the same molecule[47], such as complex carboxylic acids or acid anhydrides. In newly formed nano-sized diamonds, the high reactivity of surface carbon atoms could have also induced the conversion of defective sp[3]-bonded carbon into graphitic or sp[2]–sp[3] amorphous carbon surface domains[39–41,45,48] (Fig. 3c). These processes, which reduce the total surface energy, might have inhibited physical or chemical interaction with solvent hydrous fluids at lower temperature[37,45].

## Discussion
The discovery of sub-structures of carboxyl, carboxylate, and sp[3]-hybridized (CH$_x$)n groups ($x = 2, 3$) in Lago di Cignana diamonds is pivotal as it provides, for the first time, confirmatory evidence that carboxylic acids can be stable and quantitatively significant in hydrous fluids at pressures of 3 GPa[12]. At mantle conditions, the geosynthesis of organic molecules is expected to be thermodynamically favored by the gradients of pH and redox

surface species and the development of the reactions on pristine diamond surfaces. Previous studies reveal that nanodiamonds contain a large number of dangling carbon atoms at the surface, depending on surface characteristics, crystallographic faces, and density of impurities[36,37]. As a consequence, the surfaces are highly unstable and adsorb a variety of chemical groups (e.g.,

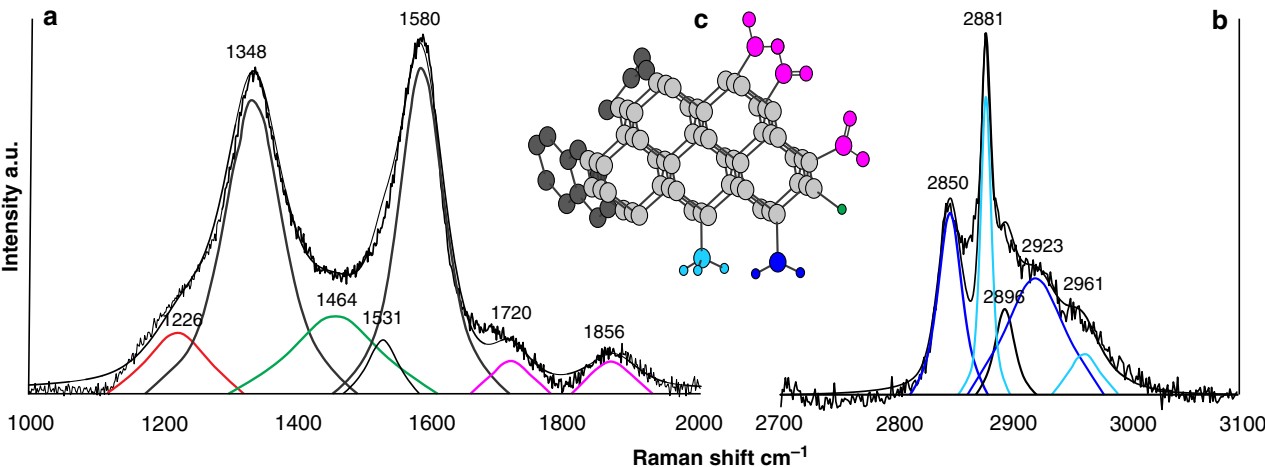

**Fig. 3** Raman spectrum showing surface organic functional groups in diamonds. **a** The Raman spectrum is showing surface functional group vibrations in a nano-sized diamond (red band profile at 1226 cm$^{-1}$) covered by a shell of sp$^2$-bonded carbon. COO$^-$ groups are revealed by two broad bands centered at around 1720 and 1856 cm$^{-1}$ (pink band profiles). **b** The bands lying from 2800 to 3000 cm$^{-1}$ correspond to C-H vibrations in (CH$_2$)$_n$ and (CH$_3$)$_n$ groups (dark blue and light blue band profiles, respectively). The fitted band at 2896 cm$^{-1}$ (black band profile) corresponds to both (CH$_2$)$_n$ and (CH$_3$)$_n$ groups vibrations. **c** A schematic model showing the structure of nano-sized diamond surfaces revealed by Raman spectra. The crystalline diamond surface is covered by sp$^2$-bonded carbon and organic functional groups

potential[11,12] that prevail in the Lago di Cignana metamorphic units. Carbon saturation could have been catalyzed when hydrous fluids started to equilibrate with the buffering slab-rock minerals (variation of $f_{O_2}^{fluid/rock}$ is buffered by EMOD [enstatite + magnesite][forsterite + diamond] equilibrium[9,20]). Therefore, the present results suggest that high-pressure decomposition of carboxylic acids induced carbon saturation in fluids by chemoselective reduction[49] of the carboxylate group and oxidation of Fe$^{2+}$ in the rock (e.g., garnet), such as:

$$R - COOH + 3H^+ + 2Fe^{2+} \rightarrow R\bullet + C + 2Fe^{3+} + 2H_2O,$$
$$(1)$$

where $R$ is any number of sp$^3$-hybridized (CH$_x$)$_n$ groups ($x = 2$, 3), and the approximate oxidation state of carbon in carboxylate groups is assumed to be +2 (ref. [49]). Such a model mass-balance reaction does not exclude that diamonds might have also formed from aqueous inorganic carbon[9]. However, carbon in organic molecules should be more readily reduced than it would be carbon bonded in CO$_2$-bearing ions or molecules, which is more oxidized and of lower chemical potential[12].

One physical model for diamond nucleation and growth at the $P–T$ conditions of Lago di Cignana UHP metamorphic complex ($P \geq 3.2$ GPa; $T = 600$ °C), consistent with present observations, is schematically illustrated in Fig. 4. In deep hydrous fluids, the carboxylate groups from organic acids could have released the initial $C^0$ in a bond-breaking event induced by fluid/rock reaction (Eq. 1) (stage I in Fig. 4). It can be assumed that diamond nuclei formed in precursor amorphous—presumably hydrogenated—carbon clusters with the structure of mixed sp$^3$ + sp$^2$ bonding. As the concentration of carbon atoms in the fluid gradually increased, the conversion of precursor phases was possibly favored by tetrahedral sp$^3$-bonded (CH$_x$)$_n$ structural groups acting as a template, leading to the formation of diamond-structured carbon lattices[50,51]. At this initial step of nucleation (stage II in Fig. 4), the presence of hydrogen-terminated radicals could have been relevant to stabilize dangling bonds of carbon on the surface of C particles, hence facilitating further sp$^3$-structured carbon growth. These processes led to the crystallization of nano-sized diamonds still containing a relevant fraction of disordered to amorphous sp$^2$

and sp$^3$ carbon and organic functional groups (stage III in Fig. 4), in some cases along with other diamond polytypes (Fig. 2e). This stage of growth completeness could have rearranged, probably by progressive desorption of H, to form crystalline diamonds up to a few micrometers in size with a shell of sp$^2$-bonded carbon, often terminated by *trans*-(CH)$_n$ (stage IV in Fig. 4).

The formation mechanism for crustal diamonds that emerges from the present study is consistent with nucleation models via metastable molecular precursors recently proposed for nano-crystal synthesis[51–54]. In particular, a nucleation pathway with tetrahedral H-terminated C groups acting as templates, before forming the sp$^3$-structured C phase, resonates with Gebbie et al.[51] experiments of diamond gas synthesis using diamondoid molecules as protonuclei. Their observation that diamond nucleation occurred through metastable, hydrogen-terminated diamond-like carbon clusters (e.g., pentamantane) led to propose a different molecular nucleation mechanism where the nucleation energetic barrier is orders of magnitude lower than that generally assumed by classical models[51]. The similarity with laboratory synthesis seemingly supports that a part of crustal diamonds may have nucleated at pressures outside the actual stability field of diamond[55]. Data do not suggest a shallow origin for diamonds, however. The growth of diamond and non-diamond phase mixtures up to a few micrometers across still requires high pressure and temperature conditions.

Pinning down the diversity of geochemical processes, which lead to the transport, or fixation (as diamond), of carbon at mantle conditions is understandably extremely challenging. Present results highlight the wide variety of soluble carbon species that can be present in deep aqueous fluids at subduction zones. It is generally accepted that subduction diamonds form through reduction of CO$_2$ and carbonate ions present in hydrous media[7–9]. Present results prove that $P$, $T$, and pH conditions, occurring during high-pressure fluid/rock redox evolution, promote the formation of organic molecules[11,12], in the absence of biologically catalyzed processes[56]. Given the chemical reactions and physical processes proposed for diamond formation, the presence of carbon–hydrogen intermediates would appear as a needed requirement to provide templates facilitating nucleation and growth.

**Table 2 Band-fitting parameters as derived from Raman spectra of carbon-based phases measured in diamonds inside fluid inclusions**

| Sample no. | FI no. | Diamond (cm⁻¹) | | | sp²-disordered C (cm⁻¹) | | | | sp²–sp³ a-C (cm⁻¹) | Transpoly acetylene groups (cm⁻¹) | sp³ ($CH_x$)n, Methylene, Methyl groups (cm⁻¹) | | | Carboxyl, Carboxylate groups (cm⁻¹) |
|---|---|---|---|---|---|---|---|---|---|---|---|---|---|---|
| | | Cubic | Defect | Met. | D | FWHM | G | FWHM | a-C | $CH_n$ | $CH_2$ | $CH_2/CH_3$ | $CH_3$ | C=O in acid and anhydride |
| M2B | B3 | | 1214 | | 1349 | 180 | 1577 | 82 | 1522 | | | | | 1862 |
| M2B | B5 | | 1223 | | 1351 | 226 | 1576 | 87 | 1525 | | | | | 1856 |
| D2AG | B1 | 1331 | 1224 | | 1365 | 134 | 1579 | 87 | 1524 | | | | | |
| G2C | I10 | | 1220 | | 1378 | 174 | 1594 | 98 | 1520 | | | | | 1728 |
| G2C | D1 | | 1222 | | 1364 | 167 | 1591 | 80 | 1525 | | | | | 1720 |
| G2C | E1 | | 1226 | | 1348 | 94 | 1580 | 77 | 1531 | 1464 | | | | |
| G2C | C1 | | 1223 | | 1364 | 168 | 1591 | 80 | 1526 | | | | | 1723 |
| ALC2 | B2 | 1328 | | 1284 / 1301 | 1361 | 60.5 | 1586 | 16 | 1566 | 1440 / 1159 | 2849 / 2847 | 2896 | 2881 / 2880.6 | |
| ALC2 | B5 | 1330 | | | 1365 | 34 | v.w. | | 1565 | | | | | |
| ALC2 | B6 | 1329 | | 1283 / 1299 | 1360 | 68 | 1588 | 11 | 1565 | 1478 / 1161 | 2850 / 2923 | 2897 | 2881 / 2961 | |
| M2B | C7 | 1332 | | | 1368 | 79 | 1599 | 56 | 1530 | 1451 / 1162 | | | | |
| G2C | D5 | 1334 | | | 1374 | 70.2 | 1590 | 87 | | 1443 / 1154 | | | | |
| G2C | E4a | 1331 | | | 1375 | 57 | 1600 | 50 | | 1443 | | | | |
| G2C | E4b | 1333 | | | 1378 | 38 | 1583 | | | 1443 | | | | |
| D2AG | B1c | 1332 | | | 1346 | 80 | 1583 | 69 | | | | | | |
| ALC2 | 1 | 1330 | | | 1364 | 89 | 1584 | 80 | | | | | | |
| M2B | C1 | 1330 | | | 1374 | 112 | 1583 | 93 | | | | | | |
| M2B | C3 | 1333 | 1220 | | 1363 | 110 | 1589 | 68 | | | | | | |
| 3D2AG | A1 | 1328 | | | 1368 | 90.2 | 1564 | 64 | | | | | | |
| D2AGB1 | E1 | 1329 | | | 1370 | 90.6 | 1565 | 73 | | | | | | |

*FI fluid inclusion, FWHM full-width at half-maximum height, met. metastable, a-C amorphous C, v.w. very weak*

---

**Table 3 Calculated fraction of disordered sp²-structured carbon in diamonds**

| Sample no. | FI no. | sp² C fraction (%) sp²/sp² + sp³ |
|---|---|---|
| M2B | C1 | 0.11 |
| M2B | C3 | 2.28 |
| D2AG | B1 | 8.63 |
| D2AGB1 | E1 | 0.30 |
| G2C | D1 | 6.66 |
| M2B | C7 | 0.71 |
| ALC2 | B6 | 3.63 |
| G2C | 5D | 0.09 |
| G2C | E4 | 0.05 |
| D2AGB | 1C | 4.90 |
| 3D2AG | A1 | 0.11 |

*FI fluid inclusion*

Taken together, the results presented in this research show bridge-building pathways between geochemistry and organic geochemistry. Miller[57] first reported organic compounds produced by abiotic synthesis under Primitive-Earth atmosphere conditions. More recently, studies of chemical reactions, which could have been significant for the prebiotic evolution on Earth, have been mostly based on organic compounds and water present in hydrothermal vents at oceanic spreading centers, and meteorites[58–62] (i.e., carbonaceous chondrites). For example, plausible models have been proposed where exogenous organic molecules were delivered intact to Earth, during the period of heavy bombardment from about 3.8 to 4.5 Gy ago[63,64]. Thus, the discovery of organic molecules, where carbon is bonded by oxygen and hydrogen, in deep hydrous fluids can constitute a relevant—presently unconsidered—carbon reservoir in the Earth's mantle. It remains unknown whether organic molecules might ultimately lead to sugars and other building blocks of life at these extreme conditions. Nonetheless, the diamond formation pathway revealed by the present study adds a new endogenous perspective to the appearance of the simplest forms of bio-chemical compounds in the Earth.

## Methods

**Sample preparation**. Five double-polished rock sections of about 100 µm thickness were selected for the Raman study of carbon-based phases contained inside unopened fluid inclusions located at 10 to 20 µm depth below the sample surface. Using well-established techniques, the rock section polishing was performed with alumina paste. Textural features of the host rock and fluid inclusions were characterized using a petrographic microscope.

**Raman microspectroscopy**. Microstructural and chemical in situ analysis of carbon-based phases inside aqueous fluid inclusions was performed using a Labram confocal Raman microscope (Horiba Scientific, Japan). The polarized Raman spectra were excited using two green Ar-ion lasers operating at 514 and 532 nm, respectively. Raman scattering excited by visible light is more sensitive to π-bonds formed by sp²-hybridized orbitals. Thus, using green laser lines, sp² carbon cross-section results about 55 times more intense than that of sp³-bonded carbon[65]. I did not use a ultraviolet (UV) laser line (244 nm), which provides similar excitation for both sp² and sp³ carbon bonds, because of the insufficient penetration of UV laser below the sample surface.

Raman spectra acquisition was performed with a backscattered geometry by focusing the laser beam inside fluid inclusions to a depth of 20 µm below the sample surface by means of a transmitted light Olympus B40 microscope. A ×100 objective (numerical aperture [NA] = 0.90) with a long-working distance was used for all the acquisitions to increase spatial resolution (<1 µm) allowing analyses of sub-micrometer phases. Raman scattering intensity of an individual crystalline phase depends on its volume, and decreases of orders of magnitude when crystals have sizes smaller than the laser spot size. In order to analyze nanometer-sized diamonds (<1 µm), confocal geometry of the Raman system was maintained under 1 µm (x–y direction) and 2 µm (z-direction) to allow spatially resolved analyses, with significant reduction, or absence, of scattering from the host garnet, and liquid

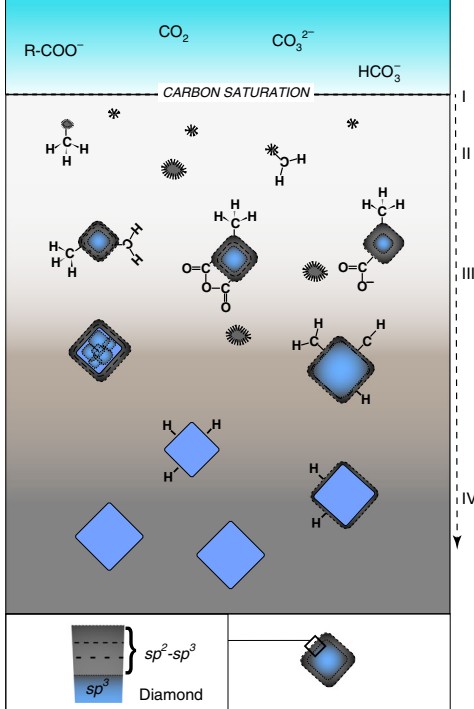

**Fig. 4** Schematic formation pathway for diamonds from dissolved organic and inorganic carbon compounds in deep hydrous fluids. (I) Carbon saturation ($C^0$) on the reduction of carboxyl-containing groups from dissolved organic acids and formation of disordered $sp^2$-, $sp^3$-bonded amorphous carbon as an intermediate state preceding the crystallization of diamond. (II) Early stages of nucleation of crystalline diamonds facilitated by tetrahedrally coordinated radical structural intermediates. (III) Further growth brings to nano-sized diamonds coated by disordered $sp^2$ carbon, and amorphous $sp^2$, $sp^3$ carbon. Metastable diamond polytypes can also form. (IV) Crystalline microdiamonds with almost no $sp^2$-bonded disordered carbon

and solid phases contained in fluid inclusions (see Fig. 2), depending on their scattering properties.

The maximum intensity of Raman diamond band was obtained from the focal spot position in the middle of the crystal. With the pinhole between 50 and 100 µm, the Raman vibrational modes of diamond completely disappeared at focal spot locations more than 2 to 3 µm away, along the $x–y$ direction, depending on diamond size. For nano-sized diamonds with weak Raman scattering, spectra were collected for variable acquisition times (from 5 to 180 s) and 1 to 3 acquisitions to get the best signal-to-noise ratio[36]. In order to prevent decomposition of structural groups eventually present on diamond surfaces, the laser power has been maintained low, generally below 50–70 mW. The spectrometer was calibrated using a natural diamond standard. The optimization of the spectral resolution was achieved by varying the slit width from 100 to 150 µm. Band central position accuracy of 1.5 cm$^{-1}$ was obtained by combining a 600 g/mm diffraction grating and the long focal length of the spectrograph at 700 mm.

**Statistical analysis of Raman spectra**. Raman spectra were baseline corrected and processed by statistical analysis (Fityk software; ref. [66]) The envelope of the whole spectrum was deconvoluted into single-band profiles using a Voigt pseudo function, which is a convolution of a Lorentzian with a Gaussian line shape[67]. In that way, the central position of the bands is defined with an accuracy better than 0.2 cm$^{-1}$, and the integrated intensity and width of single bands are adapted to the mixed spectrum. Selected Raman spectra are shown in Figs. 2, 3 and detailed information about parameters collected for each Raman band analyzed (e.g., intensity, area, and width) can be found in Tables 1–3.

When diamond spectra showed $sp^2$-bonded carbon vibrations, the $sp^2$ carbon fraction was calculated as

$$\frac{sp^2}{sp^2+sp^3} = \left(\frac{I_G}{(55 \times I_D)+I_G}\right) \times 100\#, \qquad (2)$$

where $I_G$ is the intensity of G band within the range 1560–1590 cm$^{-1}$, and $I_D$ is the intensity of the diamond peak at 1332 cm$^{-1}$ corrected for the weaker scattering at the considered laser lines[65,68].

## Data availability
The source data are provided in Tables 1–3 and in Supplementary Table 1. Raw Raman spectra are available from the author upon reasonable request.

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

## Acknowledgements

I am grateful to Marco Orlandi for insightful discussions on organic redox reactions. This research was supported by the Italian MIUR: (1) PRIN grant (no. 2017LMNLAW) and (2) Progetto Dipartimenti di Eccellenza 2018–2022. Raman facilities were provided by the Dipartimento di Scienze dell'Ambiente e della Terra, Università Milano-Bicocca.

## Author contributions

M.L.F. solely performed analyses and contributed to the writing of this article.

## Competing interests

The author declares no competing interests.
