## [Peer Review File · Nature Communications]

Reviewers' comments:

Reviewer #1 (Remarks to the Author):

Review of Frezzotti

The nature of the C-species from which diamonds have formed has long involved a debate in the literature between those who favor reduction of CO₂ and those who favor oxidation of CH₄. However, recent theoretical models have been used to suggest diamond formation may have involved aqueous organic carbon species. For example, species with intermediate oxidation states between CO₂ and CH₄ may also be important during water-rock interactions - particularly species such as formate, acetate, and propionate. This manuscript presents the first strong evidence of the association of organic carboxylate species and natural diamonds. The author has described microdiamond inclusions inside fluid inclusions in garnet crystals. These results are important because they are direct evidence that diamond formation reactions are much more complicated than traditionally depicted. The research presented is an analytical tour de force - to have studied these tiny inclusions within inclusions. The manuscript is very clearly written. I have only a few minor comments that could be addressed by the author:

lines 123-125: Not clear to me what you are saying here. Does this sentence mean that no free water molecules were detected? And, related to this, from the first part of the sentence ("Although diamonds are contained inside aqueous solution..."), how do you know this if no free water was detected?

lines 146-147: Reaction 1 is a possibility if R = H, i.e. HCOO⁻ (formate), in which the C-oxidation state is +II.

lines 146-147: What evidence is there that Fe²⁺ has been oxidized to Fe³⁺?

lines 146-147: If R = CH₃, i.e. CH₃COO⁻ (acetate), the C-oxidation state is 0, which is the same as in diamond, and no redox change is needed. In other words, you wouldn't need Fe²⁺ in the reaction.

If R is a longer chain than acetate, e.g. propionate, the C-oxidation state is < 0.

In that case, the C- in the fluid has to become oxidized to get to diamond, not reduced. You could then use a reaction with, say propionate and CO_3^{--} on the left-hand side to give diamond. In this case also you would not need to use Fe^{2+} .

lines 189-190: the bridge is really between geochemistry and organic geochemistry.

lines 190-192: The organic synthesis literature is not only on meteorites. There is also a substantial literature involving the mid-ocean ridge hydrothermal systems. And there is always the spark discharge gang (Miller-Bada-Cleaves and so on).

Overall, I recommend acceptance of the manuscript with very minor revisions.

Reviewer #2 (Remarks to the Author):

Review of manuscript by Frezzotti "Diamond growth from organic compounds..."

The paper discusses organic compounds found around microdiamonds in garnets of the Lago di Cignana UHP. The author proposes that decomposition of carboxylic acids induces diamond nucleation while tetrahedral C-H radicals create a 3D framework facilitating diamond growth. The author suggests that the organic compounds at ~100 km depth could be one of prebiotic forms of life deep in the Earth interior. My opinion of the paper is very high and I'd like to see this study published, but major revision is necessary to strengthen the arguments.

I would like the revised manuscript to address the following points:

1. The evidence for the H₂O-rich fluid looks unconvincing to me. Why the fluid was interpreted as water-rich if OH⁻ and H₂O bands were not observed in the Raman data? I dug further into the references supporting the argument and discovered that the hydrous-rich nature of the fluid in the Lago di Cignana fluid inclusions was suggested in the author's older papers (Frezzotti et al., 2011; 2014) on the basis of 1) detection of HCO₃⁻ species in solution, 2) the absence of significant molecular CO₂ in the vapor; 3) the consensus in the literature that H₂O-enriched nature of crustal

fluids responsible for diamond formation in UHP in general; 4) modelling of the fluid equilibrated at EMOG at the redox state of the Lago di Cignana rocks. The evidence seems weak. Firstly, the author does not explain why water from the allegedly H₂O-rich fluid has not been detected in the Raman or IR analyses. Water is readily seen in the Raman and IR spectrometry of fluids associated with other diamond types, like fibrous diamonds (e.g. Weiss, Y., Kiflawi, I., Navon, O., 2010. IR spectroscopy: quantitative determination of the mineralogy and bulk composition of fluid microinclusions in diamonds. *Chemical Geology* 275 (1–2), 26–34). Secondly, the modelling of the fluid which shows it to be with XH₂O = 0.992 – 0.997 (Frezzotti et al., 2014) was carried out on the assumption that the fluid was buffered by peridotitic minerals (EMOD buffer). This is a wrong assumption and the wrong paragenesis for the quartz-spessartine rock of the Lago di Cignana; even eclogite buffers for diamond-bearing parageneses of Luth (Luth R. W., 1993, *Diamonds, eclogites, and the oxidation state of the Earth's mantle. Science* 261, 66–68) would be more appropriate. Please provide additional evidence for the hydrous character of the diamond-precipitating fluid.

2. The association of organic compounds with diamond is explained as quenching of the diamond formation where the product (diamond) is seen together with reactants (organic compounds). This crucial point is the foundation of the second part of the manuscript on the control of the tetrahedral C-H radicals on the growth of the diamond lattice. I would like to see a stronger argument that the organic compounds are not products of the diamond decomposition. . The author reports “a decrease of diamond size ... with increasing amorphous and disordered sp²-bonded carbon content” and “anti-correlation between the intensities of crystalline diamond peaks and those of the intermediate products in Raman spectra”, which are compatible with the diamond replacement by hydrocarbons. In Lago di Cignana rocks, like in all metamorphic rocks, the evidence for the retrograde reactions are more common than the relics of the earlier stages of prograde metamorphism. From this general rule of thumb, we are more likely to see secondary minerals and fluids replacing peak metamorphic minerals like diamond. In fact, the author in 2014 explained the Lago di Cignana assemblage of diamond with organic compounds as “traces of carbon hydrogenation in grain boundaries, as proposed by Ferrari and Robertson (2001). Since carbon hydrogenation is observed only in diamonds contained within fluid inclusions, these data may suggest incipient diamond dissolution in aqueous fluids (Frezzotti et al., 2014).” What new data prompted the author to change the older interpretation to the opposite?

3. Following on the previous point, the Lago di Cignana organic compounds should not necessarily be equilibrated at the same peak P-Ts as the diamonds. The statement that the organic group structures, including carboxyl, carboxylate, methyl, and methylene formed at the peak metamorphic conditions (32 kb and 600°C), like the diamonds, is an assumption. No proof has been provided for pressures and temperatures of formation of the organic compounds beyond a general possibility that they can be stable in the diamond-bearing mantle (papers by Swerjensky). It is known that hydrocarbons and deuterium daughter minerals in fluid inclusions in fibrous diamonds can form down to 200 °C and 0.2 GPa, and coexist with the deeper daughter minerals that crystallized in the diamond stability field (7GPa, 950°C) (Kopylova et al., *Carbonatitic mineralogy of natural diamond-*

forming fluids, 2010). Please show more evidence that the observed organic groups and molecules are indeed “deep”.

4. I am puzzled why the studied fluid and mineral inclusion phases do not show the displacement of Raman peaks due to the relatively high remnant pressure of 3.2 GPa. If the pressure is not holding in the host garnet, this suggests a high possibility of retrogression of the original deep mineralogy and favours the secondary speciation of the organic compounds with respect to the diamond. Any comment?

Minor points:

Lines 22-23. “The present study gives evidence that fluid/rock equilibria can provide favorable conditions for the origins of prebiotic forms of life deep in the Earth interior”. Change to “The present study gives evidence that fluid/rock equilibria can provide favorable conditions for the origins of prebiotic organic compounds deep in the Earth interior”.

Line 20. Change “induce” to “may induce”

Line 28. Change “Diamonds are tiny amounts of carbon from the Earth’s interior” to “Diamonds account for tiny amount of carbon from the Earth’s interior”

I hope my review will help to sharpen the data interpretation and the paper will be a valuable contribution to our understanding of the carbon behavior in the deep Earth interior.

M. Kopylova

22 April, 2019

Review #3 (Remarks to the Author):

Review – Diamond growth from organic compounds in hydrous fluids deep within the Earth.

Prof. Frezzotti is presenting spectroscopic evidence for organic compound functional groups (Carboxyl, carboxylate, methyl and methylene) adsorbed to metamorphic microdiamonds found in fluid inclusions within garnets. Organic compounds have been previously suspected to be important for diamond-genesis and the deep carbon cycle, but direct evidence is scarce. Accordingly, Prof. Frezzotti discusses her exciting observations in an overall comprehensive way and presents a significant contribution to our current understanding of the complexity, synthesis and distribution of carbon species under the extreme conditions of the Earth's interior. The presented spectroscopic data is generally compelling, especially considering the challenges of micro-analysis of such small samples. The presented data gives room for discussion and thought experiments and encourages the search for organic compounds in the similar and other HP-HT geological settings worldwide.

The interpretation of the spectroscopic data is very interesting, and certainly a model that is worth to present and discuss. Yet, I think, that parts of the discussion are a bit too speculative and require a little more back up from literature (see below). The biggest issue is the assumption that the fluid inclusion chemistry is “quenched” and is therefore reflecting the organic chemistry that was present at the time of diamond-precipitation, i.e. peak metamorphism. Although I think that dissolved organic compounds (whether ultimately organic or abiogenic as created by “geosynthesis”) can be important for diamond formation, the study misses out to discuss the possibility of organic compound generation on the retrograde PT-path of the diamond-bearing metamorphic complex. I am convinced that a discussion of this alternative would strongly strengthen the manuscript and would (even more) highlight the importance of the generation and presence of more organic molecules in deep Earth systems.

Listed below are the most important remarks on the manuscript that I think are necessary/important to address, followed then by a list of some minor issues and comments. If these points are addressed adequately, it would be nice to see this manuscript published in *Nature Communications*.

With kind regards,
Nico Kueter

Institute of Geochemistry and Petrology
Federal Institute of Technology (ETH) Zurich
Clausiusstrasse 25
8092 Zürich, Switzerland
+41 44 632 37 43

Main Issues (References cited are listed at the end of the Document)

- 1) In line 129, the Author states that: “*Present results clearly show that diamond precipitation inside the fluid inclusion was rapidly quenched allowing to preserve in the aqueous fluids a range of intermediate carbon forms.*” Unfortunately, it becomes not clear why the Fluid Inclusion (FI) and its content was quenched rapidly.

There is no constraint in the data that would point to a rapid quench of the FI and stabilization of organic compounds. Using the presence of organic compounds as indicator for rapid quenching by itself creates a circular argument.

The FI are from a metamorphic terrane that has been exhumed slowly (compared to mantle xenoliths delivered by kimberlites or similar) over the duration of few million years (e.g. Groppo et al. 2009). This certainly does not rapidly quench the content of the FI, but rather allows back reactions within the FI along the retrograde PT-path with the potential for the formation and increased adsorption of organic compounds to the diamond surface:

Nanodiamonds itself can create a catalytic surface that promotes the synthesis of organic compounds (Lin et al 2018). Nano-diamonds and presumably irregular diamond surfaces can “graphitize” or (better) “amorphize”, creating sp^3/sp^2 -hybride carbon-materials that can yield, under presence of heteroatoms, a high density of functional/catalytic sites. Synthetic surface graphitization of diamond (to produce carbon onions) is commonly performed at higher T’s (e.g. Kuznetsov & Butenko, 2005), but graphene- and amorphous carbon-like sites can form already at lower T’s somewhere between 600 - 800 °C (Zeiger et al. 2016; Cebik et al 2013), creating a highly disordered layer around the diamond nuclei. Considering the protracted, slow ascent from peak-metamorphic conditions to the surface (A path likely to be outside the diamond stability field), it is not unlikely that diamond surfaces developed at least partially graphitic/amorphous (sp^2/sp^3 -hybride-) layers, as actually evidenced in the Raman spectra. In fact, graphitization of natural nano- & microdiamonds has been reported (see the nice discussion on graphite and other carbon-materials in diamond-bearing inclusions from Greece by Perraki et al 2006).

Considering now the complex inclusion composition (COH-fluid species, some carbonates, salts [e.g. Line 77]), there is ample opportunity to form organic compounds at lower temperatures, either by catalysis of the hybridized diamond-surface (Lin et al. 2018) or among the fluid constituents (e.g. COH-species and salts/dissolved metals functioning as catalyzer). In the latter case, (nano-) diamond surfaces are very susceptible to adsorb organic functional groups at lower temperatures (e.g. Kuznetsov & Butenko, 2005).

→In fact, the model proposed in Figure 4 could be read in the opposite way, then explaining the retrograde synthesis of complex organic compounds by the surface amorphization (activation) of nano and micro-diamond. This would be an exciting real-world example displaying the complex chemistry between amorphous carbon, diamond and organic compounds, beautifully bridging Earth Science, Organic Chemistry and Material Sciences.

- 2) Paragraph line 153 to 169. Although an interesting idea is presented, I think it is too speculative in its current way written, especially since other retrograde reactions are possible (see point 1). Another point:

Line 160 onward: “As the concentration of carbon atoms in the fluid gradually increased, the conversion of precursor phases was possibly favored by tetrahedral (CH_x)_n structural groups acting as a template, leading to the growth of diamond-structured carbon”. → This is quite a broad claim that requires some literature backup. E.g. Lambrecht et al. (1993) notes the importance of the hydrogenation of graphitic precursors to form diamond nuclei. One may also mention diamondoid-like precursor molecules as adamantane as “*template*”, see for example Matsumoto & Matsui (1983).

- 3) Line 113-114: “Noteworthy, none of the analyzed Raman bands correspond to metamorphic graphite.” → Why exactly? Because the second-order graphite peaks are missing (as shown in the spectra in Beyssac et al, 2002)? That sentence needs a bit clarification, and a reference to the Supplement S2 would be helpful.
- 4) Line 104-106 “...nano-sized diamond spectra...”: Figure 2a and 3 are supposed to show a nanodiamond. It is really not clear to me, how nanodiamond can be inferred if there is no spectral evidence (i.e. asymmetric peak at c. 1325 cm⁻¹). There are indeed experiments showing that spectra of synthetic nanodiamond can be hidden under an amorphous carbon layer (Molachin et al. 2009), but in this case one may argue that there never was a diamond in the first place. The spectrum could be simply amorphous carbon, leftovers from former organic material. Further, in the supplement S2 it is stated, that the broad band at c. 1220 cm⁻¹ may be attributed to nanodiamond, though disputed. In our organic decomposition experiments (far away from the diamond stability field) we find a similar peak at the same position (as may be inferred from Fig3, spectrum 2 in Kueter et al 2019). In fact, the D4 defect band is located at this position and is typically observed for thermally decomposed organic material (see Sadezky et al., 2005). One may check if possible, whether the D4 gives an overtone in the second order region (c. 2450 cm⁻¹). If not, this may then actually hint towards nano-sized diamond, if I am not mistaken. I suggest to better clarify the indicators for nano-diamond, or point out more specifically why they are suspected although the spectra shown in Fig 2a and 3 are not conclusive.

Minor issues and suggestions:

- 1) Figure 1b is not addressed in the main text.
- 2) Sentence in line 53-55 (“*Compared to...*”): The last part of sentence is difficult to understand. Maybe 2 separate sentences would help.
- 3) Line 98-99 (“*A subgroup...*”): The sentence is unclear to me. What subgroup and what exactly does the percentage mean and what is its relevance? A reference to the supplement S2 would be helpful.
- 4) Line 99 – 103: This sentence is difficult to understand and might miss a comma: “... *diamond peak is shifted [...] actively, suppressed and broadened.*”. Actively shifting sounds strange as well, who is pushing?
- 5) More out of curiosity, Line 123-125 (“*Although diamonds...*”): If diamonds are mostly hosted in in aqueous solution, why isn’t there any signal of the O-H, shouldn’t there be at least some signal even if you focus into the diamond? For nanodiamonds, water could actually contribute to a peak-broadening of the G-peak towards higher wave numbers, then centering around 1640 cm⁻¹ (Mochalin et al. 2009).
- 6) Line 131 “*The presence of dangling bonds at the surface and of abundant defects was previously reported in crustal microdiamonds¹⁹[...]*”. Citation 19 is a review on microdiamonds by Dobrzhinetskaya (2012), in which I could not find a statement about dangling bonds on diamond surfaces. Maybe check the reference again?
- 7) Figure 3 is really nice and well thought-out, but the color coding is very difficult to distinguish: The dark and bright blue for the C-H compounds are hard to distinguish in both, printout and on screen. I suspect there is also a color coding in red for the C-O bands and atoms, but I cannot distinguish it. I am aware that color coding is always annoying to do, but since it is such a nice figure, it would be very helpful to increase the color contrast or change colors. Also maybe label the dangling organic compounds and the graphene layer on the nano-diamond model?

Literature cited:

Beysac, O., Goffé, B., Chopin, C., & Rouzaud, J. N. (2002). Raman spectra of carbonaceous material in metasediments: a new geothermometer. *Journal of metamorphic Geology*, 20(9), 859-871.

Cebik, J., McDonough, J. K., Peerally, F., Medrano, R., Neitzel, I., Gogotsi, Y., & Osswald, S. (2013). Raman spectroscopy study of the nanodiamond-to-carbon onion transformation. *Nanotechnology*, 24(20), 205703.

Dobrzhinetskaya, L. F. (2012). Microdiamonds—Frontier of ultrahigh-pressure metamorphism: A review. *Gondwana Research*, 21(1), 207-223.

Groppo, C., Beltrando, M., & Compagnoni, R. (2009). The P–T path of the ultra-high pressure Lago Di Cignana and adjoining high-pressure meta-ophiolitic units: insights into the evolution of the subducting Tethyan slab. *Journal of Metamorphic Geology*, 27(3), 207-231.

Kuznetsov, V. L., & Butenko, Y. V. (2005). Nanodiamond graphitization and properties of onion-like carbon. In *Synthesis, properties and applications of ultrananocrystalline diamond* (pp. 199-216). Springer, Dordrecht.

Kueter, N., Lilley, M. D., Schmidt, M. W., & Bernasconi, S. M. (2019). Experimental carbonatite/graphite carbon isotope fractionation and carbonate/graphite geothermometry. *Geochimica et Cosmochimica Acta*.

Lambrecht, W. R., Lee, C. H., Segall, B., Angus, J. C., Li, Z., & Sunkara, M. (1993). Diamond nucleation by hydrogenation of the edges of graphitic precursors. *Nature*, *364*(6438), 607.

Lin, Y., Sun, X., Su, D. S., Centi, G., & Perathoner, S. (2018). Catalysis by hybrid sp²/sp³ nanodiamonds and their role in the design of advanced nanocarbon materials. *Chemical Society Reviews*, *47*(22), 8438-8473.

Matsumoto, S., & Matsui, Y. (1983). Electron microscopic observation of diamond particles grown from the vapour phase. *Journal of Materials Science*, *18*(6), 1785-1793.

Mochalin, V., Osswald, S., & Gogotsi, Y. (2009). Contribution of functional groups to the Raman spectrum of nanodiamond powders. *Chemistry of Materials*, *21*(2), 273-279.

Perraki, M., Proyer, A., Mposkos, E., Kaindl, R., & Hoinkes, G. (2006). Raman micro-spectroscopy on diamond, graphite and other carbon polymorphs from the ultrahigh-pressure metamorphic Kimi Complex of the Rhodope Metamorphic Province, NE Greece. *Earth and Planetary Science Letters*, *241*(3-4), 672-685.

Sadezky, A., Muckenhuber, H., Grothe, H., Niessner, R., & Pöschl, U. (2005). Raman microspectroscopy of soot and related carbonaceous materials: spectral analysis and structural information. *Carbon*, *43*(8), 1731-1742.

Zeiger, M., Jäckel, N., Mochalin, V. N., & Presser, V. (2016). carbon onions for electrochemical energy storage. *Journal of Materials Chemistry A*, *4*(9), 3172-3196

Reviewers' comments:

Answers to the reviewers in blue

Reviewer #1 (Remarks to the Author):

Review of Frezzotti

The nature of the C-species from which diamonds have formed has long involved a debate in the literature between those who favor reduction of CO₂ and those who favor oxidation of CH₄. However, recent theoretical models have been used to suggest diamond formation may have involved aqueous organic carbon species. For example, species with intermediate oxidation states between CO₂ and CH₄ may also be important during water-rock interactions - particularly species such as formate, acetate, and propionate. This manuscript presents the first strong evidence of the association of organic carboxylate species and natural diamonds. The author has described microdiamond inclusions inside fluid inclusions in garnet crystals. These results are important because they are direct evidence that diamond formation reactions are much more complicated than traditionally depicted. The research presented is an analytical tour de force - to have studied these tiny inclusions within inclusions. The manuscript is very clearly written. I have only a few minor comments that could be addressed by the author:

lines 123-125: Not clear to me what you are saying here. Does this sentence mean that no free water molecules were detected? And, related to this, from the first part of the sentence ("Although diamonds are contained inside aqueous solution..."), how do you know this if no free water was detected?

*I want to thank the reviewer for bringing my attention to this inconsistency in the manuscript. I have rephrased the sentence which reads now: *Hydration and hydroxylation on the surface have not been observed, since O-H stretching (3200 - 3800 cm⁻¹) and bending (1650 cm⁻¹) water bands^{35,36} are absent in the diamond spectra (Fig. 3).**

When analyzing diamonds inside fluid inclusions, no OH⁻ or H₂O Raman vibrations are observed on the surfaces. Confocal geometry of the Raman system allows a significant reduction of scattering from the host mineral phase and from liquid water and daughter phases in the inclusions, which is very beneficial for collecting in-situ data from micro- and nano-sized diamonds. In order to detect functional groups on the surface of nano and micro-sized diamonds in a fluid inclusion inside a mineral, the instrumental confocality of the Raman microspectrometer should be maintained with a spatial resolution in the 1 μm range. Thus, Raman analyses of diamonds do not bring information on the composition of enclosing fluids inside inclusions. I have added some more information on the Raman analytical set up in the Methods section.

Fluid inclusions are water-dominated. It can be seen by the microscope (see also micro-photographs in Fig. 1).

I have done additional microthermometric analyses of some fluid inclusions in order to measure the ice melting temperatures and have some new data on the salinity of this aqueous fluid, reported in the Supplementary Information section.

lines 146-147: Reaction 1 is a possibility if R = H, i.e. HCOO⁻ (formate), in which the C-oxidation state is +II.

Concerning redox reactions involving organic molecules, I have preferred to follow the organic chemistry approach (in particular ref. 50), which is distinct from that of inorganic chemistry, where redox reactions are defined as gain/loss of electrons and decrease/increase of oxidation number of the whole molecule. According to March (ref. 50), the practice in organic chemistry is to set up a series of functional groups in a qualitative way and then to define reduction the conversion of one functional group in a molecule of a lower category.

I have tried to make this approach clearer by rewriting the formula giving more emphasis to functional groups.

lines 146-147: What evidence is there that Fe²⁺ has been oxidized to Fe³⁺?

The Fe³⁺/Fe_{tot} ratio in minerals was calculated by stoichiometry in ref. 9. This paper also reports WDS chemical maps of garnet showing enrichment of Fe in areas containing fluid inclusions.

lines 146-147: If R = CH₃⁻, i.e. CH₃COO⁻ (acetate), the C-oxidation state is 0, which is the same as in diamond, and no redox change is needed. In other words, you wouldn't need Fe²⁺ in the reaction.

If R is a longer chain than acetate, e.g. propionate, the C-oxidation state is < 0.

In that case, the C- in the fluid has to become oxidized to get to diamond, not reduced. You could then use a reaction with, say propionate and CO₃²⁻ on the left-hand side to give diamond. In this case also you would not need to use Fe²⁺.

See above comments to line 146-147.

lines 189-190: the bridge is really between geochemistry and organic geochemistry.

Corrected.

lines 190-192: The organic synthesis literature is not only on meteorites. There is also a substantial literature involving the mid-ocean ridge hydrothermal systems. And there is always the spark discharge gang (Miller-Bada-Cleaves and so on).

Right! I have added it in text along with relevant references (refs.58-61).

Overall, I recommend acceptance of the manuscript with very minor revisions.

I am extremely thankful for the reviewer suggestions. I am delighted for the very positive feedback, especially, on the significance and the interest of present study on the genesis of crustal diamonds, and that he/she recommends the publication of my study in Nature Communications after very minor revisions.

Reviewer #2 (Remarks to the Author):

Review of manuscript by Frezzotti "Diamond growth from organic compounds..."

The paper discusses organic compounds found around microdiamonds in garnets of the Lago di Cignana UHP. The author proposes that decomposition of carboxylic acids induces diamond nucleation while tetrahedral C-H radicals create a 3D framework facilitating diamond growth. The author suggests that the organic compounds at ~100 km depth could be one of prebiotic forms of life deep in the Earth interior. My opinion of the paper is very high and I'd like to see this study published, but major revision is necessary to strengthen the arguments.

I am particularly grateful for the reviewer's high evaluation of the paper and her comments and suggestions. I agree with the reviewer that the text needs improvement. I have modified the text following the reviewer's suggestions and considering all the comments raised. Point to point answers are reported below.

I would like the revised manuscript to address the following points:

The evidence for the H₂O-rich fluid looks unconvincing to me. Why the fluid was interpreted as water-rich if OH- and H₂O bands were not observed in the Raman data? I dug further into the references supporting the argument and discovered that the hydrous-rich nature of the fluid in the Lago di Cignana fluid inclusions was suggested in the author's older papers (Frezzotti et al., 2011; 2014) on the basis of 1) detection of HCO₃⁻ species in solution, 2) the absence of significant molecular CO₂ in the vapor; 3) the consensus in the literature that H₂O-enriched nature of crustal fluids responsible for diamond formation in UHP in general; 4) modelling of the fluid equilibrated at EMOG at the redox state of the Lago di Cignana rocks. The evidence seems weak. Firstly, the author does not explain why water from the allegedly H₂O-rich fluid has not been detected in the Raman or IR analyses. Water is readily seen in the Raman and IR spectrometry of fluids associated with other diamond types, like fibrous diamonds (e.g. Weiss, Y., Kiflawi, I., Navon, O., 2010. IR spectroscopy: quantitative determination of the mineralogy and bulk composition of fluid microinclusions in diamonds. *Chemical Geology* 275 (1–2), 26–34).

I agree with the reviewer that the sentence is unclear. What I was trying to write was that, when analyzing diamonds inside fluid inclusions, OH- or H₂O Raman vibrations are not observed on the surfaces. The sentence has been rewritten "*Hydration and hydroxylation on the surface have not been observed since O-H stretching (3200 - 3800 cm⁻¹) and bending (1650 cm⁻¹) water bands^{35,36} are absent in the diamond spectra (Fig. 3)*".

Fluid inclusions are aqueous. It can be seen optically at the microscope (see also microphotographs in Fig. 1). The 4 points listed by the reviewer are based on the observation of water inside inclusions, not the other way around. Since inclusions are aqueous, we analyzed the liquid water by Raman to detect dissolved species (point 1 and 2; refs. 9 and 21). Since inclusions are aqueous, we modeled the diamond-forming C-O-H fluid speciation at the considered P-T-fO₂ conditions (point 4; ref.21). In this case, results confirmed previous findings showing that subduction diamonds could form at the water maximum (point 3; ref. 21).

Raman analysis allows to distinguish the different phases optically and to focus on a single phase (solid or liquid or gas). For present work, a confocal set up of the Raman system allows a significant reduction of scattering from the host mineral phase and from liquid water and daughter phases in the inclusions, which is very beneficial for collecting in-situ data from micro- and nano-sized diamonds. Thus, the spectra of diamond do not bring information on the composition of the fluid contained inside the inclusion.

In this revised version of the manuscript, I have added new microthermometric analyses of some fluid inclusions in order to measure the ice melting temperatures and have some new data on the salinity of this aqueous fluid. Results are reported in the Supplementary Information section.

In order to give more explicit information on the analytical setup, I have detailed the analytical conditions necessary to analyze diamonds which have dimensions similar or smaller than the laser spot size in the Methods section.

Secondly, the modelling of the fluid which shows it to be with $X_{H_2O} = 0.992 - 0.997$ (Frezzotti et al., 2014) was carried out on the assumption that the fluid was buffered by peridotitic minerals (EMOD buffer). This is a wrong assumption and the wrong paragenesis for the quartz-spessartine rock of the Lago di Cignana; even eclogite buffers for diamond-bearing parageneses of Luth (Luth R. W., 1993, Diamonds, eclogites, and the oxidation state of the Earth's mantle. *Science* 261, 66–68) would be more appropriate. Please provide additional evidence for the hydrous character of the diamond-precipitating fluid.

I agree with the reviewer that the sentence in parenthesis is unclear. I have rewritten the sentence as *“variation of fluid/rock f_{O_2} is buffered by EMOD equilibrium”*.

The calculation of fluid/rock redox equilibrium of Lago di Cignana metasediments containing diamond has been addressed in two previous manuscripts (refs. 9 and 21). The oxygen fugacity was calculated by thermodynamic modeling of mineral phases in studied rocks. PERPLEX calculations indicate that diamond and carbonate in studied metasediments can coexist at oxygen fugacities between FMQ-2 and FMQ +1.5 at the conditions of UHP metamorphism recorded at Lago di Cignana of 600°C and 3.4 GPa (ref. 9; Fig. 3 in the supplementary material of ref. 9). Since the fluid is H_2O rich, Gfluid further allows calculating f_{O_2} between FMQ-1 and FMQ+08, at the considered P-T conditions (ref. 21). Although f_{O_2} values are near those of FMQ, it does not imply that the variation of f_{O_2} must be parallel to the FMQ equilibrium. It could also be buffered by an EMOD equilibrium: EMOD-1.2 and EMOD 0.6 at peak metamorphism. The detailed report of all the calculations can be found in ref. 21.

The association of organic compounds with diamond is explained as quenching of the diamond formation where the product (diamond) is seen together with reactants (organic compounds). This crucial point is the foundation of the second part of the manuscript on the control of the tetrahedral C-H radicals on the growth of the diamond lattice. I would like to see a stronger argument that the organic compounds are not products of the diamond decomposition. .

I agree with the reviewer that the origin of surface functional groups was not sufficiently discussed in the text, although considered. In the revised version of the manuscript, I have added a chapter on surface modification and reactivity in nano-sized diamond where the possible origin of O and

H-bonded organic functional groups and sp^2 carbon on surfaces are discussed in detail. Relevant literature has been reported as well.

In this chapter, the Raman data are interpreted in order to attribute the origin of organic moieties. Nanodiamond contains a large number of dangling carbon atoms at the surfaces. As a consequence, these adsorb a variety of chemical groups (e.g., carboxyl, anhydride, lactone, methyl, methylene) from the crystallization medium, and reconstruct as (more stable) amorphous or graphitic C. For this reason, they show complex structures characterized by an sp^3 -C core and an sp^2 -carbon surface with adsorbed functional groups (cf., refs. 37-39).

In nanodiamonds, exposure to distinct solvents is generally performed at lower temperature to selectively eliminate part of the surface species, depending on the required properties of functionalized nanodiamonds (cf., refs. 37-47).

1) Thermal annealing in vacuum, H_2 , He or N_2 is performed to produce nanodiamonds characterized by an sp^3 -bonded carbon core and an sp^2 amorphous (or graphitic) carbon shell (e.g., material sciences applications). Thermal annealing allows increasing the sp^2 carbon fraction and H-terminated carbon groups while removing O-bonded carbon species.

2) Oxidation in air, CO_2 , or aqueous phase is performed to increase the possibility of adsorbing functional groups and molecules on the surfaces (e.g., biomedical applications) by removing the “stable” sp^2 -C shell. This process increases the content of oxygen-containing functional groups (e.g., O-H, C=O, COOH), and induces adsorption of hydroxyl groups on surfaces. By removal of the sp^2 -shell, the nanodiamond overall size decreases.

In studied diamonds, none of reported evolutions is observed. In addition, being the diamonds contained in aqueous fluids, the latter surface evolution should be expected. However, carboxyl and carboxylate groups ($\cong 1700-1850\text{ cm}^{-1}$) are observed only in those diamonds with the highest H- and sp^2 -bonded carbon bands ($\cong 1300-1600\text{ cm}^{-1}$) (Fig. 2a and 3a). This suggests that (C=O)-containing surface species cannot result from oxidation in aqueous fluids. Moreover, bands arising from O-H groups ($\cong 1650\text{ cm}^{-1}$), which form by reaction with or on adsorption from water, are absent in spectra (Figs. 2 and 3a).

Thus, based on present analyses, I conclude that the organic O and H-bearing functional groups on the surfaces of Lago di Cignana diamonds were generated by adsorption or reaction with dissolved chemical species in the crystallization fluids at conditions relevant to deep subduction zone environment.

The author reports “a decrease of diamond size ... with increasing amorphous and disordered sp^2 -bonded carbon content” and “anti-correlation between the intensities of crystalline diamond peaks and those of the intermediate products in Raman spectra”, which are compatible with the diamond replacement by hydrocarbons.

What I intended to say is that they are smaller in size. I have eliminated the word “decrease”, which suggests a dynamic evolution which cannot be inferred from spectra. I have also eliminated the word anti-correlation (substituted with variable sp^2/sp^3 carbon ratio), for the same reason.

Bulk dissolution of diamonds in aqueous fluids at high temperature releases mainly methane and CO_2 +/- H_2 . The typically proposed reaction is $\text{C} + \text{H}_2\text{O} = \text{CO}_2 + \text{CH}_4$. These gaseous species are absent in the fluid inclusions.

In Lago di Cignana rocks, like in all metamorphic rocks, the evidence for the retrograde reactions are more common than the relics of the earlier stages of prograde metamorphism. From this general rule of thumb, we are more likely to see secondary minerals and fluids replacing peak metamorphic minerals like diamond.

I agree with the reviewer that these processes are commonly observed in rocks. However, analyzed diamonds are contained within primary fluid inclusions in garnet, indicating trapping at peak metamorphism conditions. In addition, in order to be preserved on decompression to the surface, fluid inclusions should remain isolated (i.e., closed). If the water reacts with host garnet or is lost during decompression, the fluid inclusions would show re-equilibration features or would be empty (ref. 18).

In fact, the author in 2014 explained the Lago di Cignana assemblage of diamond with organic compounds as "traces of carbon hydrogenation in grain boundaries, as proposed by Ferrari and Robertson (2001). Since carbon hydrogenation is observed only in diamonds contained within fluid inclusions, these data may suggest incipient diamond dissolution in aqueous fluids (Frezzotti et al., 2014).

" What new data prompted the author to change the older interpretation to the opposite?

Hydrogen, previously detected on diamond surfaces (ref. 21), was a first indication of surface reactivity in diamonds. Trans-polyacetylene chains were not recognized as organic moieties (ref. 21). They were considered as H-terminated surface C, following Ferrari and Robertson's (ref. 24) which first attributed the 1100 and 1400 cm^{-1} bands to trans-(CH)_n. Previously, these bands were considered as ultra-nanodiamonds vibrations.

The interpretation of the Raman data collected during the present study made me further understand diamond surface reconstruction processes. In the revised manuscript, I have tried to make clear on which observations the conclusion that adsorption of organic moieties occurred at crystallization conditions is based.

Present Raman data suggest that organic functional groups and sp^2 -bonded carbon clusters formed by pristine surface reactivity at crystallization conditions. Late reaction with inclusion water at low temperature (dissolution or oxidation) does not seem a relevant process, since it would have induced the formation of O- and O-H bonded functional groups on the diamond surfaces and removed the sp^2 carbon shell. This is not observed in Raman spectra, where C=O containing groups are observed along with H-terminated organic carbon groups and amorphous sp^2 carbon. This excludes oxidation in water, as also suggested by the absence of OH-containing functional groups on surfaces.

Following on the previous point, the Lago di Cignana organic compounds should not necessarily be equilibrated at the same peak P-Ts as the diamonds. The statement that the organic group structures, including carboxyl, carboxylate, methyl, and methylene formed at the peak metamorphic conditions (32 kb and 600°C), like the diamonds, is an assumption. No proof has

been provided for pressures and temperatures of formation of the organic compounds beyond a general possibility that they can be stable in the diamond-bearing mantle (papers by Swerjensky).

I would respectfully disagree with the reviewer concerning the DEW model by Dimitri Sverjensky. His theoretical model allows predicting carbon speciation not only in terms of gaseous species (CO_2 , CH_4), but also, as CO_3^{2-} , HCO_3^- , and organic molecules in H_2O . I think that his theoretical model has profoundly impacted our view of C-O-H fluids at depth. Quite some new research on diamond formation and Earth's Carbon cycle stemmed from his model.

Present findings give confirmatory evidence that organic acids can be stable at extreme P and T in aqueous fluids, as predicted by the DEW model.

It is known that hydrocarbons and deuteric daughter minerals in fluid inclusions in fibrous diamonds can form down to 200 °C and 0.2 GPa, and coexist with the deeper daughter minerals that crystallized in the diamond stability field (7GPa, 950°C) (Kopylova et al., Carbonatitic mineralogy of natural diamond-forming fluids, 2010). Please show more evidence that the observed organic groups and molecules are indeed "deep".

I agree with the reviewer on her comments on the composition of fluid inclusions in diamonds. Concerning daughter minerals, being these formed by solute supersaturation on cooling, they can precipitate at distinct P-T conditions, depending on single species solubility and concentration. Diamond formed at peak UHP metamorphic conditions, however.

As discussed above, in the revised version of the manuscript, I have added a chapter on surface modification and reactivity in diamond where the possible origin of O and H-bonded organic functional groups and sp^2 carbon on surfaces are discussed in detail (see answers above).

I am puzzled why the studied fluid and mineral inclusion phases do not show the displacement of Raman peaks due to the relatively high remnant pressure of 3.2 GPa. If the pressure is not holding in the host garnet, this suggests a high possibility of retrogression of the original deep mineralogy and favours the secondary speciation of the organic compounds with respect to the diamond. Any comment?

Mineral inclusions in peak phases at high P-T conditions do show overpressure remnants (i.e., band upshift) in Raman spectra.

The present Raman work focusses, however, on daughter mineral phases which are included in aqueous fluids trapped as inclusions. Inside these microsystems, the pressure is regulated by the fluid equation of state and is function of trapping T conditions (isochoric fluid inclusion behavior). Thus, fluid inclusion the recorded pressure at laboratory conditions is low. For this reason, daughter minerals in liquid water do not show Raman band shifting.

Significant fluid overpressure was recorded by the inclusions during decompression at high temperature from peak metamorphic conditions, when the pressure recorded by the fluid in the inclusions was much higher than the lithostatic one. Indeed, several inclusions show Raman evidence for stretching (i.e., an increase of the fluid inclusion volume revealed by plastic deformation in surrounding garnet host), which would further decrease the density of the fluid, as reported in the Supplementary Information section. In the revised version of the manuscript, I have tried to clarify this point.

Minor points:

Lines 22-23. "The present study gives evidence that fluid/rock equilibria can provide favorable conditions for the origins of prebiotic forms of life deep in the Earth interior". Change to "The present study gives evidence that fluid/rock equilibria can provide favorable conditions for the origins of prebiotic organic compounds deep in the Earth interior".

Corrected

Line 20. Change "induce" to "may induce"

Corrected

Line 28. Change "Diamonds are tiny amounts of carbon from the Earth's interior" to "Diamonds account for tiny amount of carbon from the Earth's interior"

Corrected

I hope my review will help to sharpen the data interpretation and the paper will be a valuable contribution to our understanding of the carbon behavior in the deep Earth interior.

M. Kopylova

22 April, 2019

Reviewer #3 (Remarks to the Author)

Review – Diamond growth from organic compounds in hydrous fluids deep within the Earth.

Prof. Frezzotti is presenting spectroscopic evidence for organic compound functional groups (Carboxyl, carboxylate, methyl and methylene) adsorbed to metamorphic microdiamonds found in fluid inclusions within garnets. Organic compounds have been previously suspected to be important for diamond-genesis and the deep carbon cycle, but direct evidence is scarce. Accordingly, Prof. Frezzotti discusses her exciting observations in an overall comprehensive way and presents a significant contribution to our current understanding of the complexity, synthesis and distribution of carbon species under the extreme conditions of the Earth's interior. The presented spectroscopic data is generally compelling, especially considering the challenges of micro-analysis of such small samples. The presented data gives room for discussion and thought experiments and encourages the search for organic compounds in the similar and other HP-HT geological settings worldwide.

The interpretation of the spectroscopic data is very interesting, and certainly a model that is worth to present and discuss. Yet, I think, that parts of the discussion are a bit too speculative and require a little more back up from literature (see below). The biggest issue is the assumption that the fluid inclusion chemistry is "quenched" and is therefore reflecting the organic chemistry that was present at the time of diamond-precipitation, i.e. peak metamorphism. Although I think that dissolved organic compounds (whether ultimately organic or abiogenic as created by "geosynthesis") can be important for diamond formation, the study misses out to discuss the

possibility of organic compound generation on the retrograde PT-path of the diamond-bearing metamorphic complex. I am convinced that a discussion of this alternative would strongly strengthen the manuscript and would (even more) highlight the importance of the generation and presence of more organic molecules in deep Earth systems.

Listed below are the most important remarks on the manuscript that I think are necessary/important to address, followed then by a list of some minor issues and comments. If these points are addressed adequately, it would be nice to see this manuscript published in Nature Communications.

With kind regards,

Nico Kueter

I highly appreciate the most insightful comments and suggestions provided by the reviewer. I agree with the reviewer on the importance of organic molecules in deep aqueous fluids. I also fully agree that the origin of non-diamond carbon and functional groups on diamond surfaces should have been discussed in much more detail in the text.

In the corrected version of the manuscript, I have tried to make clear the diamond surface evolution adding a chapter in the Result section. In this chapter, the Raman data are interpreted in order to attribute the origin of organic moieties and sp²-amorphous and graphitic carbon detected on surfaces. Point to point answers are reported below.

Institute of Geochemistry and Petrology
Federal Institute of Technology (ETH) Zurich
Clausiusstrasse 25
8092 Zürich, Switzerland
+41 44 632 37 43

Main Issues (References cited are listed at the end of the Document)

In line 129, the Author states that: "Present results clearly show that diamond precipitation inside the fluid inclusion was rapidly quenched allowing to preserve in the aqueous fluids a range of intermediate carbon forms." Unfortunately, it becomes not clear why the Fluid Inclusion (FI) and its content was quenched rapidly. There is no constraint in the data that would point to a rapid quench of the FI and stabilization of organic compounds. Using the presence of organic compounds as indicator for rapid quenching by itself creates a circular argument. The FI are from a metamorphic terrane that has been exhumed slowly (compared to mantle xenoliths delivered by kimberlites or similar) over the duration of few million years (e.g. Groppo et al. 2009). This

certainly does not rapidly quench the content of the FI, but rather allows back reactions within the FI along the retrograde PT-path with the potential for the formation and increased adsorption of organic compounds to the diamond surface.

I want to thank the reviewer for bringing my attention to this inconsistency. What I intended to say is that diamonds preserved the functional groups attached to pristine surfaces at high temperatures and did not react further. In the light of the reviewer comment is also evident, however, that a more detailed interpretation of the Raman features from diamond surfaces should have been added in the text.

This sentence has been eliminated. In the corrected version of the manuscript, I have added a chapter to discuss the structural features of diamond surfaces and the origin of functional groups. See, the more detailed answer to the next point.

Nanodiamonds itself can create a catalytic surface that promotes the synthesis of organic compounds (Lin et al 2018). Nano-diamonds and presumably irregular diamond surfaces can “graphitize” or (better) “amorphize”, creating sp^3/sp^2 -hybride carbon-materials that can yield, under presence of heteroatoms, a high density of functional/catalytic sites. Synthetic surface graphitization of diamond (to produce carbon onions) is commonly performed at higher T's (e.g. Kuznetsov & Butenko, 2005), but graphene- and amorphous carbon-like sites can form already at lower T's somewhere between 600 – 800 °C (Zeiger et al. 2016; Cebik et al 2013), creating a highly disordered layer around the diamond nuclei. Considering the protracted, slow ascent from peak-metamorphic conditions to the surface (A path likely to be outside the diamond stability field), it is not unlikely that diamond surfaces developed at least partially graphitic/amorphous (sp^2/sp^3 -hybride-) layers, as actually evidenced in the Raman spectra. In fact, graphitization of natural nano- & microdiamonds has been reported (see the nice discussion on graphite and other carbon-materials in diamond-bearing inclusions from Greece by Perraki et al 2006).

Nanodiamonds contain a large number of dangling carbon atoms at the surfaces. As a consequence, these adsorb a variety of chemical groups (e.g., carboxyl, anhydride, lactone, methyl, methylene) from the crystallization medium, and reconstruct as (more stable) sp^2 (or sp^2/sp^3 hybrid) amorphous or graphitic C. For this reason, they show complex structures characterized by an sp^3 -C core and an sp^2 -carbon surface with adsorbed functional groups.

The point raised is if “graphitization” in Lago di Cignana diamonds occurred at lower temperature: that it is the product of surface reaction with the enclosing medium at a later stage.

In synthetic diamonds, exposure to distinct solvents at lower temperature is generally performed to selectively eliminate part of the surface species, depending on the required properties of functionalized nanodiamond.

Thermal annealing in vacuum, H_2 , He or N_2 is performed to produce nanodiamonds characterized by an sp^3 -bonded carbon core and an sp^2 amorphous (or graphitic) carbon shell (e.g., material sciences applications). Thermal annealing allows increasing the sp^2 carbon fraction and H-terminated carbon groups while removing O-bonded carbon species.

Conversely, oxidation in air, CO_2 , or aqueous phase is performed to increases the possibility of adsorbing functional groups and molecules on the surfaces (e.g., biomedical applications) by

removing the “stable” sp^2 -C shell. This process increases the content of oxygen-containing functional groups (e.g., O-H, C=O, COOH), and induces adsorption of hydroxyl groups on surfaces. By removal of the sp^2 -shell, the nanodiamond overall size decreases.

In studied diamonds, none of these two evolutions is observed. In particular, being the diamond included in aqueous fluids, the latter surface evolution should be expected, more than the first one. However, carboxyl and carboxylate groups ($\cong 1700$ - 1850 cm^{-1}) are observed only in those diamonds with the highest H- and sp^2 -bonded carbon bands ($\cong 1300$ - 1600 cm^{-1}) (Fig. 2a and 3a). This suggests that “graphitization” cannot be the relevant process.

Based on present analyses, I conclude that the organic O and H-bearing functional groups on the surfaces of Lago di Cignana diamonds were generated by adsorption or reaction with dissolved chemical species in the crystallization fluids at conditions relevant to deep subduction zone environment.

Considering now the complex inclusion composition (COH-fluid species, some carbonates, salts [e.g. Line 77]), there is ample opportunity to form organic compounds at lower temperatures, either by catalysis of the hybridized diamond-surface (Lin et al. 2018) or among the fluid constituents (e.g. COHspecies and salts/dissolved metals functioning as catalyzer). In the latter case, (nano-) diamond surfaces are very susceptible to adsorb organic functional groups at lower temperatures (e.g. Kuznetsov & Butenko, 2005).

As discussed above, transformations of structures and chemistry of diamond surface by exposure to a fluid/gas medium, are induced either by surface oxidation or by reduction, depending on the nature enclosing gas/fluid medium. Purification of synthetic nanodiamonds in O_2 -air, CO_2 or water brings to an increase of C=O containing groups. It is performed to eliminate the amorphous sp^2/sp^3 domains also typically found in nanodiamonds formed by detonation or CVD processes (broad sense). In Lago di Cignana diamonds both oxidized and reduced species coexist on the surfaces, while O-H groups are absent. These pieces of evidence support preservation of species adsorbed at formation conditions and exclude late oxidation in aqueous fluids.

In fact, the model proposed in Figure 4 could be read in the opposite way, then explaining the retrograde synthesis of complex organic compounds by the surface amorphization (activation) of nano and micro-diamond. This would be an exciting real-world example displaying the complex chemistry between amorphous carbon, diamond and organic compounds, beautifully bridging Earth Science, Organic Chemistry and Material Sciences.

The reviewer concern is entirely understandable, considering the text in the initially submitted manuscript which did not report a detailed interpretation of diamond surface features.

As discussed above, In the corrected version of the manuscript I have tried to make clear the reasons why the functional groups on diamond surfaces cannot be a late process, by adding a chapter in the Result section.

In my opinion, however, the proposed model cannot be readily interpreted oppositely. To see the evolution of diamond surfaces oppositely, in fact, one should imagine that nanodiamonds progressively form a thicker amorphous sp^2 carbon shell. Following experimental work, the sp^2 -

carbon/diamond ratio increases during thermal annealing in hydrogen at increasing temperature, not at decreasing temperature.

2) Paragraph line 153 to 169. Although an interesting idea is presented, I think it is too speculative in its current way written, especially since other retrograde reactions are possible (see point 1).

Another point:

Line 160 onward: "As the concentration of carbon atoms in the fluid gradually increased, the conversion of precursor phases was possibly favored by tetrahedral (CH_x)_n structural groups acting as a template, leading to the growth of diamondstructured carbon". This is quite a broad claim that requires some literature backup. E.g. Lambrecht et al. (1993) notes the importance of the hydrogenation of graphitic precursors to form diamond nuclei. One may also mention diamondoid-like precursor molecules as adamantane as "template", see for example Matsumoto & Matsui (1983).

I thank the reviewer for highlighting the weakness of a model proposed in the absence of relevant literature. The proposed model is grounded on the classical model of diamond growth by May (ref. 51). The formation mechanism that emerges from the present study can find an explanation in recent nucleation models via metastable molecular precursors, proposed for nanocrystal synthesis (e.g., refs. 52,53,54,55).

Further, I agree that the mechanisms of diamondoids growth are central. In particular, a nucleation pathway with tetrahedral H-terminated C groups acting as templates, before forming the sp³-structured C phase, shows striking similarity with recent synthesis experiments by Gebbie et al. (ref. 52), where diamond nucleation occurred through metastable, hydrogen-terminated diamond-like carbon clusters (e.g., pentamantane). The authors proposed a molecular nucleation mechanism different from that generally assumed by classical models, where the nucleation energetic barrier is orders of magnitude lower than generally considered for bulk sp³ carbon.

These concepts are now reported in text, along with relevant literature. My proposal of a physical model for diamond nucleation and growth could help to better understand a process that is generally considered in terms of redox reactions, or thermodynamic equilibrium between carbon allotropes. How does low-temperature crustal diamond nucleation occur? Certainly, nucleation via a metastable structural precursor represents an exciting hypothesis to be further explored.

3) Line 113-114: "Noteworthy, none of the analyzed Raman bands correspond to metamorphic graphite." Why exactly? Because the second-order graphite peaks are missing (as shown in the spectra in Beyssac et al, 2002)? That sentence needs a bit clarification, and a reference to the Supplement S2 would be helpful.

The spectral features of sp²-bonded carbon (e.g., band position and relative FWHM) follow the amorphization trajectory of Ferrari and Robertson (ref. 24). This is discussed in the Supplementary Information section.

4) Line 104-106 "...nano-sized diamond spectra...": Figure 2a and 3 are supposed to show a nanodiamond. It is really not clear to me, how nanodiamond can be inferred if there is no spectral

evidence (i.e. asymmetric peak at c. 1325 cm^{-1}). There are indeed experiments showing that spectra of synthetic nanodiamond can be hidden under an amorphous carbon layer (Molachin et al. 2009), but in this case one may argue that there never was a diamond in the first place. The spectrum could be simply amorphous carbon, leftovers from former organic material. Further, in the supplement S2 it is stated, that the broad band at c. 1220 cm^{-1} may be attributed to nanodiamond, though disputed.

In our organic decomposition experiments (far away from the diamond stability field) we find a similar peak at the same position (as may be inferred from Fig3, spectrum 2 in Kueter et al 2019). In fact, the D4 defect band is located at this position and is typically observed for thermally decomposed organic material (see Sadezky et al., 2005). One may check if possible, whether the D4 gives an overtone in the second order region (c. 2450 cm^{-1}). If not, this may then actually hint towards nano-sized diamond, if I am not mistaken. I suggest to better clarify the indicators for nano-diamond, or point out more specifically why they are suspected although the spectra shown in Fig 2a and 3 are not conclusive.

The fitted 1220 cm^{-1} centered band is intense and evident in fig. 2a, b and c (with or without diamond) and in fig. 3a. A similar fitted band is usually attributed to nano-sized diamonds (ref. 28-29).

As the reviewer suggests, the absence of the overtone spectrum in the 2500 cm^{-1} region, argues against attribution to the D4 defect in graphite. In addition, in my opinion, also the fitted band shape does not suggest an attribution to the D4 defect, mainly because the intensity appears to be too high relative to the D band, and the FWHM in D4 is expected to be much broader. By looking at the other spectra in fig. 2, it is evident that the intensity of the D4 mode is very weak, and does not emerge substantially in fitted band components.

Minor issues and suggestions:

1) Figure 1b is not addressed in the main text.

Added.

2) Sentence in line 53-55 (“Compared to...”): The last part of sentence is difficult to understand. Maybe 2 separate sentences would help.

The sentence has been rewritten as two separate sentences.

3) Line 98-99 (“A subgroup...”): The sentence is unclear to me. What subgroup and what exactly does the percentage mean and what is its relevance? A reference to the supplement S2 would be helpful.

There is no relevance, indeed. I have rewritten the sentence.

4) Line 99 – 103: This sentence is difficult to understand and might miss a comma: “... diamond peak is shifted [...] actively, suppressed and broadened.”. Actively shifting sounds strange as well, who is pushing?

No one... Corrected.

5) More out of curiosity, Line 123-125 (“Although diamonds...”): If diamonds are mostly hosted in aqueous solution, why isn’t there any signal of the O-H, shouldn’t there be at least some signal even if you focus into the diamond? For nanodiamonds, water could actually contribute to a peak-broadening of the G-peak towards higher wave numbers, then centering around 1640 cm⁻¹ (Mochalin et al. 2009).

I agree with the reviewer that the common assumption that water (low Raman scattering) does not interfere with Raman spectra of nanodiamonds (extremely high Raman scattering) is not true.

I was able to avoid a liquid water contribution in spectra of diamonds keeping the confocality of the Raman spectrometer at around 1 μm³ focusing on the diamond.

For crystal sizes lower than the analyzed sample volume, the intensity of the Raman scattering is, among other parameters, a function of the relative volumes of the analyzed crystals and enclosing medium. In their paper, Mochalin et al. (2009) reported measurements of a liquid water contribution adding to the OH-groups on the surfaces of synthetic nanodiamonds of < 10 nm in size in water. The authors did not report the confocality set up of their instrument, unfortunately. However, from the magnification applied (15x objective - N.A. not reported), as a rule of thumb, more than 10 μm³ should have been analyzed. This brings to extremely high water/diamond volume ratios, making extremely reasonable a water contribution in spectra. In the present study, volumes of diamonds are hundreds of cubic nanometers, in a 1 μm³ of analyzed sample (objective is 100x with N.A.=0.9, hole 50-100). Due to the relatively much higher diamond/H₂O volume ratio, it can be indeed possible not to record O-H bands from water in spectra. A similar analytical set up is also necessary to reveal vibrations from functional groups on surfaces.

I have added some more information on the Raman analytical setup in the Methods section.

6) Line 131 “The presence of dangling bonds at the surface and of abundant defects was previously reported in crustal microdiamonds¹⁹[...]”. Citation 19 is a review on microdiamonds by Dobrzhinetskaya (2012), in which I could not find a statement about dangling bonds on diamond surfaces. Maybe check the reference again?

This sentence has been eliminated.

7) Figure 3 is really nice and well thought-out, but the color coding is very difficult to distinguish: The dark and bright blue for the C-H compounds are hard to distinguish in both, printout and on screen. I suspect there is also a color coding in red for the CO bands and atoms, but I cannot distinguish it. I am aware that color coding is always annoying to do, but since it is such a nice figure, it would be very helpful to increase the color contrast or change colors. Also maybe label the dangling organic compounds and the graphene layer on the nano-diamond model?

I have tried to improve the color-coding in figure 3.

Literature cited:

Beyssac, O., Goffé, B., Chopin, C., & Rouzaud, J. N. (2002). Raman spectra of carbonaceous material in metasediments: a new geothermometer. *Journal of metamorphic Geology*, 20(9), 859-871.

Dobrzhinetskaya, L. F. (2012). Microdiamonds—Frontier of ultrahigh-pressure metamorphism: A review. *Gondwana Research*, 21(1), 207-223.

Groppo, C., Beltrando, M., & Compagnoni, R. (2009). The P–T path of the ultra-high pressure Lago Di Cignana and adjoining high-pressure meta-ophiolitic units: insights into the evolution of the subducting Tethyan slab. *Journal of Metamorphic Geology*, 27(3), 207-231.

Kueter, N., Lilley, M. D., Schmidt, M. W., & Bernasconi, S. M. (2019). Experimental carbonatite/graphite carbon isotope fractionation and carbonate/graphite geothermometry. *Geochimica et Cosmochimica Acta*.

Lambrecht, W. R., Lee, C. H., Segall, B., Angus, J. C., Li, Z., & Sunkara, M. (1993). Diamond nucleation by hydrogenation of the edges of graphitic precursors. *Nature*, 364(6438), 607.

Lin, Y., Sun, X., Su, D. S., Centi, G., & Perathoner, S. (2018). Catalysis by hybrid sp²/sp³ nanodiamonds and their role in the design of advanced nanocarbon materials. *Chemical Society Reviews*, 47(22), 8438-8473.

Matsumoto, S., & Matsui, Y. (1983). Electron microscopic observation of diamond particles grown from the vapour phase. *Journal of Materials Science*, 18(6), 1785-1793.

Mochalin, V., Osswald, S., & Gogotsi, Y. (2009). Contribution of functional groups to the Raman spectrum of nanodiamond powders. *Chemistry of Materials*, 21(2), 273-279.

Perraki, M., Proyer, A., Mposkos, E., Kaindl, R., & Hoinkes, G. (2006). Raman microspectroscopy on diamond, graphite and other carbon polymorphs from the ultrahigh-pressure metamorphic Kimi Complex of the Rhodope Metamorphic Province, NE Greece. *Earth and Planetary Science Letters*, 241(3-4), 672-685.

Sadezky, A., Muckenhuber, H., Grothe, H., Niessner, R., & Pöschl, U. (2005). Raman microspectroscopy of soot and related carbonaceous materials: spectral analysis and structural information. *Carbon*, 43(8), 1731-1742.

Zeiger, M., Jäckel, N., Mochalin, V. N., & Presser, V. (2016). carbon onions for electrochemical energy storage. *Journal of Materials Chemistry A*, 4(9), 3172-3196

Missing cited literature added.

REVIEWERS' COMMENTS:

Reviewer #2 (Remarks to the Author):

I am satisfied with the revision of the manuscript "Diamond growth from organic compounds in hydrous fluids deep within the Earth" and recommend it for publication with only few minor corrections.

1. The most important of my corrections relates to a logical scramble that can be streamlined and can make the flow of text better. I see the manuscript's two principal conclusions as 1) first empirical evidence for the organic compounds to form deep; and 2) diamond can form from dissolved organic species. These two conclusions should be more distanced from each other, as in some sense they are opposite to each other. I find several places where the author jumps from one conclusion to another, and the clarity of the text would be served better by a clearer logical separation of the conclusion. For example, the Abstract reads: "

"I obtain direct evidence that micro- and nano-diamonds are coated by sp²-, and sp³-bonded amorphous carbon that shows Raman modes of attached organic group structures, including carboxyl, carboxylate, methyl, and methylene. Results unveil that decomposition of carboxylic acids may induce diamond nucleation on the reduction of the carboxylate groups, whereas tetrahedral C-H radicals create a template allowing nucleation and growth. The present study gives evidence that fluid/rock equilibria can provide favorable conditions for the origins of prebiotic organic compounds deep in the Earth interior."

I would exchange the second and the third sentence: "I obtain direct evidence that micro- and nano-diamonds are coated by sp²-, and sp³-bonded amorphous carbon that shows Raman modes of attached organic group structures, including carboxyl, carboxylate, methyl, and methylene. The present study gives evidence that fluid/rock equilibria can provide favorable conditions for the origins of prebiotic organic compounds deep in the Earth interior. (Deals with Conclusion 1, Evidence that organic compounds exist at great depth.) Moreover, decomposition of carboxylic acids may induce diamond nucleation on the reduction of the carboxylate groups, whereas tetrahedral C-H radicals create a template allowing nucleation and growth" (Deals with Conclusion 2, induction of diamond nucleation by scaffolding of organic molecules)

Several places would benefit from similar changes. They may seem small, but would help a great deal in clarifying the interpretation.

2. As for my comment and the author's reply on buffering the fluid by EMOD fO₂, I understand that my puzzlement is mis-addressed. The argument on the carbon saturation and diamond deposition was made in the older, 2014 paper, and the new manuscript only gives a cursory reference to the detailed calculations by Frezotti et al., 2014. However, I still cannot explain to myself how the fluid in the garnet-quartz rock can be buffered by Ol-Opx phases absent in the metasedimentary unit or in the adjacent metabasites.

The remaining suggested changes are stylistic:

. Line 29: " yet revealing their genesis represents necessary input information for understanding large-scale processes" is an awkward phrase. Could be condensed to a better phrase like "yet revealing their genesis is necessary for understanding large-scale processes ..."

. Lines 54-55 "Compared to previous experimental or theoretical studies, the analysis of fluid-containing inclusions has the advantage to examine a direct sample of diamond-forming media encapsulated within minerals" could be stylistically improved to "Compared to previous experimental or theoretical studies, the analysis of fluid-containing inclusions has the advantage of examining a direct sample of diamond-forming media encapsulated within minerals"

. Lines 65- 68: In this sentence the highlighted "it" refers to the last singular noun, i.e. to "diamond formation". Should be re-phrased as the intended meaning of "it" here is apparently "ophiolitic slice":

. "The diamond-bearing ultrahigh-pressure (UHP) ophiolitic slice of Lago di Cignana (western Alps) has several characteristics which are very appropriate to study diamond formation in subduction

zones: i) it represents a fragment of an exhumed metamorphic complex of crustal origin subducted to a depth of 100 km, or more, during Alpine oceanic subduction about 35 Ma ago"

I'm looking forward to seeing the research in print.

Regards,

M. Kopylova

Reviewer #3 (Remarks to the Author):

Dear Editor,
Dear Author,

the revised manuscript and the rebuttal addresses all important points raised in my first review. The revisions were done thoroughly and with great care. I was especially pleased to read a new chapter concerning the surface chemistry of diamonds.

In my previous review, I suggested to interpret the observations in light of diamond decay on the retrograde metamorphic path. The author addresses now in detail her reasoning behind the idea that the functional groups have been adsorbed from the crystallising medium rather than being produced by the out-of-equilibrium conversion of the diamond surface. This chapter argues well for her initial model, and provides the reader with, for this topic, important information on the surface chemistry of diamond. I think the new chapter is a big plus for the manuscript and provides a good starting point for interested readers to dive into the complex surface chemistry of diamond and graphitic carbon.

As a side note, I think we should not dismiss the retrograde formation of amorphous carbon entirely. As I found the review process & the authors response very enlightening, I would like to propose to make the reviews available to the audience (i.e. use the “Transparent peer review system“ offered by *Nature Communications*). The decision is, of course, with the Author.

From my perspective, the study addresses an important topic and the manuscript is in an overall satisfying state, comprehensive and full of information. Only very few, very minor final remarks are listed below. I sincerely enjoyed the review and rebuttal and recommend a publication in *Nature Communications*.

With kind regards,
Nico Kueter

Remarks

Line 114 – “None of the analyzed Raman bands correspond to metamorphic graphite [...] (see supplementary information)”

I really appreciate that the Author gives detailed information to this statement in the Supplementary information, but it is a little hidden/indirect: It is not clearly stated that the observed D & G band features distinguish the FI amorphous carbon from metamorphic graphite (i.e. “metamorphic graphite” not mentioned in the relevant supplementary text). I would recommend to add a (half-) sentence in the supplementary text that clarifies it. This helps the reader to link the above statement (Line 114) to the additional information given in the supplement.

Section from Line 196 – 211 – “One physical...”

It would be helpful to refer in the text to the stages I to IV displayed in figure 4.

Line 199 – “...initially C_0 in a bond-breaking...”

$C_0 \rightarrow 0$ in upper case (C^0) as it refers to the neutral charge of elemental carbon.

Labels and arrows in figure 1 are very small and should be enhanced.

REVIEWERS' COMMENTS:

Answers to the reviewers in blue

Reviewer #2 (Remarks to the Author):

I am satisfied with the revision of the manuscript “Diamond growth from organic compounds in hydrous fluids deep within the Earth” and recommend it for publication with only few minor corrections.

I thank the reviewer for the useful comments bringing important improvements to the manuscript.

1. The most important of my corrections relates to a logical scramble that can be streamlined and can make the flow of text better. I see the manuscript’s two principal conclusions as 1) first empirical evidence for the organic compounds to form deep; and 2) diamond can form from dissolved organic species. These two conclusions should be more distanced from each other, as in some sense they are opposite to each other. I find several places where the author jumps from one conclusion to another, and the clarity of the text would be served better by a clearer logical separation of the conclusion. For example, the Abstract reads: “

“I obtain direct evidence that micro- and nano-diamonds are coated by sp²-, and sp³-bonded amorphous carbon that shows Raman modes of attached organic group structures, including carboxyl, carboxylate, methyl, and methylene. Results unveil that decomposition of carboxylic acids may induce diamond nucleation on the reduction of the carboxylate groups, whereas tetrahedral C-H radicals create a template allowing nucleation and growth. The present study gives evidence that fluid/rock equilibria can provide favorable conditions for the origins of prebiotic organic compounds deep in the Earth interior.”

I would exchange the second and the third sentence: “I obtain direct evidence that micro- and nano-diamonds are coated by sp²-, and sp³-bonded amorphous carbon that shows Raman modes of attached organic group structures, including carboxyl, carboxylate, methyl, and methylene. The present study gives evidence that fluid/rock equilibria can provide favorable conditions for the origins of prebiotic organic compounds deep in the Earth interior. (Deals with Conclusion 1, Evidence that organic compounds exist at great depth.) Moreover, decomposition of carboxylic acids may induce diamond nucleation on the reduction of the carboxylate groups, whereas tetrahedral C-H radicals create a template allowing nucleation and growth” (Deals with Conclusion 2, induction of diamond nucleation by scaffolding of organic molecules) Several places would benefit from similar changes. They may seem small, but would help a great deal in clarifying the interpretation.

I thank the reviewer for pointing this out. The abstract has been in part rewritten since it was too long. In rewriting it, I have followed her suggestions.

2. As for my comment and the author’s reply on buffering the fluid by EMOD fO₂, I understand that my puzzlement is misaddressed. The argument on the carbon saturation and diamond deposition was made in the older, 2014 paper, and the new manuscript only gives a cursory reference to the detailed calculations by Frezzotti et al., 2014. However, I still cannot explain to myself how the fluid in the garnet-quartz rock can be buffered by Ol-Opx phases absent in the metasedimentary unit or in the adjacent metabasites.

Concerning diamond precipitation conditions, the oxygen fugacity was calculated by thermodynamic modeling of mineral phases in studied rocks. PERPLEX calculations indicate that diamond and carbonate in studied metasediments can coexist at oxygen fugacities between FMQ-2 and FMQ +1.5 at 600°C and 3.4 GPa. As the reviewer writes, thermodynamic modelling of redox conditions through the whole P-T path of studied rocks during active subduction has been treated in detail in two previous papers.

The remaining suggested changes are stylistic:

Line 29: “yet revealing their genesis represents necessary input information for understanding large-scale processes” is an awkward phrase. Could be condensed to a better phrase like “yet revealing their genesis is necessary for understanding large-scale processes ...”

I have changed the sentence.

Lines 54-55 “Compared to previous experimental or theoretical studies, the analysis of fluid-containing inclusions has the advantage to examine a direct sample of diamond-forming media encapsulated within minerals” could be stylistically improved to “Compared to previous experimental or theoretical studies, the analysis of fluid-containing inclusions has the advantage of examining a direct sample of diamond-forming media encapsulated within minerals”

I have changed the sentence.

. Lines 65- 68: In this sentence the highlighted “it” refers to the last singular noun, i.e. to “diamond formation”. Should be rephrased as the intended meaning of “it” here is apparently “ophiolitic slice”: “The diamond-bearing ultrahigh-pressure (UHP) ophiolitic slice of Lago di Cignana (western Alps) has several characteristics which are very appropriate to study diamond formation in subduction zones: i) it represents a fragment of an exhumed metamorphic complex of crustal origin subducted to a depth of 100 km, or more, during Alpine oceanic subduction about 35 Ma ago”

I have rewritten these sentences also according to suggestions from the Editor.

I’m looking forward to seeing the research in print.

Regards,

M. Kopylova

Reviewer 3 #2nd Review – Diamond growth from organic compounds in hydrous fluids deep within the Earth.

Dear Editor, 23.08.2019

Dear Author,

the revised manuscript and the rebuttal addresses all important points raised in my first review. The revisions were done thoroughly and with great care. I was especially pleased to

read a new chapter concerning the surface chemistry of diamonds.

In my previous review, I suggested to interpret the observations in light of diamond decay on the retrograde metamorphic path. The author addresses now in detail her reasoning behind the idea that the functional groups have been adsorbed from the crystallising medium rather than being produced by the out-of-equilibrium conversion of the diamond surface. This chapter argues well for her initial model, and provides the reader with, for this topic, important information on the surface chemistry of diamond. I think the new chapter is a big plus for the manuscript and provides a good starting point for interested readers to dive into the complex surface chemistry of diamond and graphitic carbon.

As a side note, I think we should not dismiss the retrograde formation of amorphous carbon entirely. As I found the review process & the authors response very enlightening, I would like to propose to make the reviews available to the audience (i.e. use the “Transparent peer review system“ offered by Nature Communications). The decision is, of course, with the Author.

From my perspective, the study addresses an important topic and the manuscript is in an overall satisfying state, comprehensive and full of information. Only very few, very minor final remarks are listed below. I sincerely enjoyed the review and rebuttal and recommend a publication in Nature Communications.

With kind regards,

Nico Kueter

I thank the reviewer for the detailed comments and insightful suggestions which improved significantly the impact of this study.

Remarks

Line 114 – “None of the analyzed Raman bands correspond to metamorphic graphite [...] (see supplementary information)”

I really appreciate that the Author gives detailed information to this statement in the Supplementary information, but it is a little hidden/indirect: It is not clearly stated that the observed D & G band features distinguish the FI amorphous carbon from metamorphic graphite (i.e. “metamorphic graphite” not mentioned in the relevant supplementary text). I

would recommend to add a (half-) sentence in the supplementary text that clarifies it. This helps the reader to link the above statement (Line 114) to the additional information given in the supplement.

I have added a few sentences (+ 2 references) in the Supplementary Note 2, trying to clarify why spectral features of sp²-bonded carbon do not correspond to metamorphic graphite.

Section from Line 196 – 211 – “One physical...”

It would be helpful to refer in the text to the stages I to IV displayed in figure 4.

I have referred to stages I-IV Fig. 4 in the text.

Line 199 – “...initially C0 in a bond-breaking...”

C0 à 0 in upper case (C0) as it refers to the neutral charge of elemental carbon.

I have corrected this typo.

Labels and arrows in figure 1 are very small and should be enhanced.

I have increased the size of labels and arrows in Fig. 1.